# Bilevel Optimization without Lower-Level Strong Convexity from the Hyper-Objective Perspective

## Abstract

Bilevel optimization reveals the inner structure of otherwise oblique optimization problems, such as hyperparameter tuning, neural architecture search, and meta-learning. A common goal in bilevel optimization is to find stationary points of the hyper-objective function. Although this hyper-objective approach is widely used, its theoretical properties have not been thoroughly investigated in cases where the lower-level functions lack strong convexity. This work takes a step forward when the typical lower-level strong convexity assumption is absent. Our hardness results show that bilevel optimization for general convex lower-level functions is intractable to solve. We then identify several regularity conditions of the lower-level problems that can provably confer tractability. Under these conditions, we propose the Inexact Gradient-Free Method (IGFM), which uses the Switching Gradient Method (SGM) as an efficient sub-routine, to find an approximate stationary point of the hyper-objective in polynomial time.

## 1 Introduction

The goal of bilevel optimization (BLO) is to minimize the upper-level (UL) function $f(x, y)$ under the constraint that $y$ is minimized w.r.t. the lower-level (LL) function $g(x, y)$ on a closed convex set $\mathcal{Y} \subseteq \mathbb{R}^{d_y}$. Mathematically, it can be formulated as:

$$\min_{x \in \mathbb{R}^{d_x}, y \in Y^*(x)} f(x, y), \quad Y^*(x) = \arg\min_{y \in \mathcal{Y}} g(x, y). \tag{1}$$

BLO in this form has received increasing attention due to its wide applications in many machine learning problems, including hyperparameter tuning (Franceschi et al., 2018; Pedregosa, 2016), neural architecture search (Liu et al., 2019; Wang et al., 2022b; Zoph & Le, 2016; Zhang et al., 2021), meta-learning (Franceschi et al., 2018; Hospedales et al., 2021; Ravi & Larochelle, 2017; Pham et al., 2021), out-of-distribution learning (Zhou et al., 2022), adversarial training (Goodfellow et al., 2020; Sinha et al., 2018; Lin et al., 2020a;b), reinforcement learning (Konda & Tsitsiklis, 1999; Hong et al., 2023), causal learning (Jiang & Veitch, 2022; Arjovsky et al., 2019).

The hyper-objective approaches (Dempe, 2002; Dempe & Zemkoho, 2020; Liu et al., 2020; 2021) reformulate Problem 1 by

$$\min_{x \in \mathbb{R}^d} \varphi(x), \text{ where } \varphi(x) = \min_{y \in Y^*(x)} f(x, y) \text{ is called the hyper-objective.} \tag{2}$$

It transforms the problem into the composition of a simple BLO (Sabach & Shtern, 2017) w.r.t. the LL variable $y$ and an unconstrained single-level optimization w.r.t. the UL variable $x$. This reformulation naturally leads to two foundational problems: The first one involves

*P1: Find an optimal LL variable $\hat{y} \in Y^*(\hat{x})$ such that $\varphi(\hat{x}) = f(\hat{x}, \hat{y})$ for a given $\hat{x}$.*

The second one involves

*P2: Find a UL variable $\hat{x}$ that is a stationary point of $\varphi(x)$.*

Both problems are easy to solve when the LL function is strongly convex. The lower-level strong convexity (LLSC) ensures $Y^*(x)$ to be a singleton, and therefore simplifies Equation 2 into $\varphi(x) = f(x, y^*(x))$, where the LL optimal solution $y^*(x) = \arg\min_{y \in \mathcal{Y}} g(x, y)$ can be found via gradient descent on $g$. If we further assume $\mathcal{Y} = \mathbb{R}^{d_y}$, then the implicit function theorem indicates:

$$\nabla \varphi(x) = \nabla_x f(x, y^*(x)) - \nabla^2_{xy} g(x, y^*(x))[\nabla^2_{yy} g(x, y^*(x))]^{-1} \nabla_y f(x, y^*(x)). \qquad (3)$$

Then one can apply the gradient step with $\nabla \varphi(x)$ to find a UL stationary point. This forms the basis of the classical hyper-objective approaches for BLO with LLSC (Ji et al., 2021). However, these methods heavily rely on the LLSC condition that may not hold in many applications.

This paper investigates BLO with only LL convexity, but without LLSC. Adding a regularization term to the LL function is a natural idea to ensure LLSC (Rajeswaran et al., 2019), but we show in Proposition 4.1 that any small regularization may lead to a large deviation on the hyper-objective. Furthermore, we construct hard instances to illustrate the intractability of BLO without LLSC, for both finding an LL optimal solution and a UL stationary point: Firstly, we prove a lower bound in Proposition 4.2 to show that $\varphi(x)$ is not computable in finite iterations for general convex functions. Secondly, we give a pair of $f(x, y)$ and $g(x, y)$ in Example 4.1 such that the resulting hyper-objective $\varphi(x)$ is discontinuous and thus intractable to optimize.

The constructions of these hard instances rely on the fact that a general convex LL function can be arbitrarily "flat". To avoid the intractability caused by the undesirable "flatness", we introduce two sufficient conditions that can provably confer tractability to BLO with only LL convexity: the gradient dominance condition (Assumption 5.1) and the weak sharp minimum condition (Assumption 5.2). Under these conditions, we propose novel algorithms to find an LL optimal solution and a UL stationary point, with non-asymptotic convergence guarantees:

**Finding an LL Optimal Solution.** We show that both conditions fall into a general class of the Hölderian error bound condition (Proposition G.1), under which we propose the Switching Gradient Method (SGM, Algorithm 1) to find an LL optimal solution in polynomial time (Theorem 6.1).

**Finding a UL Stationary Point.** We prove in Proposition 5.1 that both conditions imply the Lipschitz continuity of the solution mapping $Y^*(x)$, which is proved to be both sufficient and necessary for the Lipschitz continuity of $\varphi(x)$ by Proposition 4.3. Under the Lipschitz continuity of $\varphi(x)$, we then propose the Inexact Gradient-Free Method (IGFM, Algorithm 2), which can provably converge to a Goldstein stationary point (Zhang et al., 2020) of the hyper-objective by incorporating SGM as an efficient sub-routine.

We compare the intractability and tractability results under different assumptions on the LL function in Table 1, and summarize our contributions as follows:

1. We formulate the LL optimality and UL stationary as valid criteria for BLO without LLSC (Section 3), which are necessary for an optimistic optimal solution (Dempe et al., 2006).

2. We provide hardness results to show that BLO without LLSC is generally intractable. Our analysis highlights the importance of sharpness in LL functions (Section 4).

3. We prove that when the LL function satisfies either the gradient dominance condition or the weak sharp minimum condition, the hyper-objective $\varphi(x)$ is Lipschitz and thus Clarke differentiable (Section 5).

4. We propose novel polynomial time algorithms for BLO with LL convexity under either the gradient dominance or the weak sharp minimum condition (Section 6).

5. We conduct numerical experiments on adversarial training and hyperparameter tuning that showcase the superiority of our methods (Section 7).

## 2 RELATED WORKS

**BLO with LLSC.** Approximate implicit differentiation (AID) (Domke, 2012; Ghadimi & Wang, 2018; Pedregosa, 2016; Franceschi et al., 2018; Grazzi et al., 2020; Ji et al., 2021) and iterative differentiation (ITD) (Gould et al., 2016; Franceschi et al., 2017; Shaban et al., 2019; Bolte et al.,

| Assumption on LL function | LL Optimality | UL Stationary | Reference |
|---|---|---|---|
| Strongly convex | Tractable | Tractable | Known result |
| Convex with dominant gradients | Tractable | Tractable | Proved by this work |
| Convex with weak sharp minimum | Tractable | Tractable | Proved by this work |
| Only convex | Intractable | Intractable | Proved by this work |

Table 1: An overview of the theoretical results in this paper. We show that BLO without LLSC is generally intractable, but becomes tractable when the LL function satisfies either the gradient dominance or the weak sharp minimum condition.

2021) are two representative methods that have non-asymptotically convergence to a UL stationary point for BLO with LLSC. Due to their popularity, many improvements to AID and ITD have also been proposed (Chen et al., 2022; Hong et al., 2023; Yang et al., 2021; Ji & Liang, 2021; Ji et al., 2022; Dagréou et al., 2022).

**BLO without LLSC.** In the absence of LLSC, Arbel & Mairal (2022) showed that one can extend AID by replacing the inverse in Equation 3 with the Moore-Penrose inverse under the Morse-Bott condition on the manifold $\{y \in \mathbb{R}^{d_y} : \nabla_y f(x, y) = 0\}$. Liu et al. (2021; 2020) extended ITD by proposing various methods to update the LL variable. However, all the methods mentioned above are limited to asymptotic convergence to an LL optimal solution and lack analysis for finding a UL stationary point. Due to the challenge of directly optimizing the hyper-objective, some concurrent works (Liu et al., 2022; Sow et al., 2022) reformulate Problem 1 via the value-function approach and show non-asymptotic convergence to the KKT points of this equivalent problem. However, since classical constraint qualifications provably fail for the reformulated problem (Ye & Zhu, 1995), the KKT condition is even not a necessary condition for a local minimum (Example A.1). In contrast, a UL stationary point is always a necessary condition. We leave a detailed comparison of our hyper-objective approach and value-function approach in Appendix A.

## 3 PRELIMINARIES

### 3.1 NOTATIONS AND BACKGROUNDS

**Basic Notation.** Throughout this paper, we denote the LL solution mapping as $Y^*(x) = \arg\min_{y \in \mathcal{Y}} g(x, y)$, the LL value function as $g^*(x) = \min_{y \in \mathcal{Y}} g(x, y)$, and the hyper-objective as $\varphi(x) = \min_{y \in Y^*(x)} f(x, y)$. If $\varphi(x)$ has a finite minimum, we denote $\varphi^* = \inf_{x \in \mathbb{R}^{d_x}} \varphi(x)$. We use $\|\cdot\|$ to denote the $\ell_2$-norm of a vector, and $z_{[j]}$ to denote the $j$-th coordinate of vector $z$. We use $\mathbb{B}_\delta(z) = \{z' : \|z' - z\| \le \delta\}$ to denote the $\ell_2$-ball centered at $z$ with radius $\delta$. We let $\sigma_{\max}(A)$ to be the largest singular value of matrix $A$, and $\sigma_{\min}^+(A)$ to be its smallest non-zero singular value.

**Constrained Optimization.** To tackle the possible constraint in $y$, we introduce the definitions of projection and generalized gradient (Nesterov, 2018) as follows.

**Definition 3.1** (Projection). *We define the projection onto a set $\mathcal{Y}$ by $\mathcal{P}_\mathcal{Y}(\cdot) := \arg\min_{y \in \mathcal{Y}} \|y - \cdot\|$.*

**Definition 3.2** (Generalized Gradient). *For a $L$-gradient Lipschitz function $g(x, y)$ with $y \in \mathcal{Y}$, we define the generalized gradient with respect to $y$ by $\mathcal{G}_\eta(y; x) := (y - \mathcal{P}_\mathcal{Y}(y - \eta\nabla_y g(x, y)))/\eta$ with some $0 < \eta \le 1/L$.*

Note that the generalized gradient reduced to $\nabla_y g(x, y)$ when $\mathcal{Y} = \mathbb{R}^{d_y}$.

**Set-Valued Analysis.** A classic notion of distance in set-valued analysis is the Hausdorff distance (Rockafellar & Wets, 2009), formally defined as follows.

**Definition 3.3** (Hausdorff Distance). *The Hausdorff distance between two sets $S_1, S_2$ is defined as*

$$\text{dist}(S_1, S_2) = \max\left\{\sup_{x_1 \in S_1} \inf_{x_2 \in S_2} \|x_1 - x_2\|, \sup_{x_2 \in S_2} \inf_{x_1 \in S_1} \|x_1 - x_2\|\right\}$$

This allows us to define the Lipschitz continuity of set-valued mappings as follows.

**Definition 3.4.** *We call a set-valued mapping* $S(x) : \mathbb{R}^{d_1} \rightrightarrows \mathbb{R}^{d_2}$ *locally Lipschitz if for any* $x \in \mathbb{R}^{d_1}$*, there exists* $\delta > 0$ *and* $L > 0$ *such that for any* $x' \in \mathbb{R}^{d_1}$ *satisfying* $\|x' - x\| \leq \delta$*, we have* $\mathrm{dist}(S(x), S(x')) \leq L\|x - x'\|$*. We call* $S(x)$ *Lipschitz if we can let* $\delta \to \infty$*.*

Note that the above definition generalizes the Lipschitz continuity for a single-valued mapping.

**Nonsmooth Analysis.** The following Clarke subdifferential (Clarke, 1990) generalizes both the gradients of differentiable functions and the subgradients of convex functions.

**Definition 3.5** (Clarke Subdifferential). *The Clarke subdifferential of a locally Lipschitz function* $h(x) : \mathbb{R}^d \to \mathbb{R}$ *at a point* $x \in \mathbb{R}^d$ *is defined by*

$$\partial h(x) := \mathrm{Conv} \left\{ s \in \mathbb{R}^d : \exists x_k \to x, \nabla h(x_k) \to s, \text{ s.t. } \nabla h(x_k) \text{ exists for all } k \right\}.$$

It can be proved that finding a point with a small Clarke subdifferential is generally intractable for a nonsmooth nonconvex function (Zhang et al., 2020). So we need to consider the following relaxed definition of stationarity for non-asymptotic analysis in nonsmooth nonconvex optimization (Zhang et al., 2020; Tian et al., 2022; Davis et al., 2022; Jordan et al., 2023; Kornowski & Shamir, 2021; Lin et al., 2022; Cutkosky et al., 2023; Kornowski & Shamir, 2023).

**Definition 3.6** (Approximate Goldstein Stationary Point). *Given a locally Lipschitz function* $h(x) : \mathbb{R}^d \to \mathbb{R}$*, we call* $x \in \mathbb{R}^d$ *a* $(\delta, \varepsilon)$*-Goldstein stationary point if* $\min \{\|s\| : s \in \partial_\delta h(x)\} \leq \varepsilon$*, where* $\partial_\delta h(x) := \mathrm{Conv} \left\{ \cup_{x' \in \mathbb{B}_\delta(x)} \partial h(x') \right\}$ *is the Goldstein subdifferential (Goldstein, 1977).*

## 3.2 THE OPTIMALITY CONDITIONS

This section introduces the optimality conditions for BLO without LLSC used in this paper. Firstly, we recall the definition of the optimistic optimal solution (Dempe et al., 2006), which is a standard optimality condition for the hyper-objective reformulation.

**Definition 3.7.** *A pair of point* $(x^*, y^*)$ *is called a locally optimistic optimal solution to Problem 1 if* $y^* \in Y^*(x^*)$ *and there exists* $\delta > 0$ *such that we have* $\varphi(x^*) \leq \varphi(x)$ *and* $f(x^*, y^*) \leq f(x^*, y)$ *for all* $(x, y) \in \mathbb{B}_\delta(x^*, y^*)$*. It is called a globally optimistic optimal solution if we can let* $\delta \to \infty$*.*

A globally optimistic optimal solution is an exact solution to Problem 1, but its computation is NP-hard since $\varphi(x)$ is generally nonconvex (Danilova et al., 2020). A common relaxation is to find a locally optimistic optimal solution, for which we can derive the following necessary conditions.

**Proposition 3.1.** *Suppose* $f(x, \cdot)$ *and* $g(x, \cdot)$ *are convex, and* $\varphi(x)$ *is locally Lipschitz. Then for any locally optimistic optimal solution* $(x^*, y^*)$*, we have* $\partial \varphi(x^*) = 0$*,* $f(x^*, y^*) = \varphi(x^*)$ *and* $g(x^*, y^*) = g^*(x^*)$*.*

It motivates us to use the following criteria for non-asymptotic analysis:

**Definition 3.8** (UL Stationary). *Suppose* $\varphi(x)$ *is locally Lipschitz. We call* $\hat{x}$ *a* $(\delta, \varepsilon)$*-UL stationary point if it is a* $(\delta, \varepsilon)$*-Goldstein stationary point of* $\varphi(x)$*.*

**Definition 3.9** (LL Optimality). *Fix an* $x$*. Suppose* $f(x, \cdot)$ *and* $g(x, \cdot)$ *are convex. We call* $\hat{y}$ *a* $(\zeta_f, \zeta_g)$*-LL optimal solution if we have* $|f(x, \hat{y}) - \varphi(x)| \leq \zeta_f$ *and* $g(x, \hat{y}) - g^*(x) \leq \zeta_g$*.*

The main focus of this paper is to discuss when and how one can design a polynomial time algorithm to achieve the above goals for any given positive precision $\delta, \varepsilon, \zeta_f, \zeta_g$.

**Remark 3.1.** *In Definition 3.8, we assume that* $\varphi(x)$ *is locally Lipschitz, which is one of the mildest conditions to ensure Clarke differentiability. However, it may not hold for BLO without LLSC, and we will give the sufficient and necessary condition for it later in Proposition 4.3. Definition 3.8 adopts the Goldstein stationary points since* $\varphi(x)$ *can be nonconvex nonsmooth such that traditional stationary points may be intractable, as we will show later in Example 5.1.*

## 4 HARDNESS RESULTS FOR INTRACTABILITY

In this section, we provide various hardness results to show the challenges of BLO without LLSC. We first explain why one can not manually regularize the LL function to ensure the LLSC condition. Subsequently, we demonstrate that both the tasks of finding an LL optimal solution and finding a UL stationary point can be intractable for BLO without LLSC.

### 4.1 Can Regularization Help?

One natural way to tackle BLO without LLSC is to add some small quadratic terms and then apply an algorithm designed under LLSC (Rajeswaran et al., 2019). However, we show that the regularization transforms $Y^*(x)$ from a set to a singleton, thus breaking the original problem structure.

**Proposition 4.1.** *Given a pivot $\hat{y}$, there exists a BLO instance, where both $f(x, y)$ and $g(x, y)$ are convex in $y$, and the resulting hyper-objective $\varphi(x)$ is a quadratic function, but for any $\lambda > 0$ the regularized hyper-objective*

$$\varphi_\lambda(x) = \min_{y \in Y^*_\lambda(x)} f(x, y), \quad Y^*_\lambda(x) = \arg\min_{y \in \mathcal{Y}} g_\lambda(x, y) + \lambda \|y - \hat{y}\|^2$$

*is a linear function with $|\inf_{x \in \mathbb{R}^{d_x}} \varphi_\lambda(x) - \inf_{x \in \mathbb{R}^{d_x}} \varphi(x)| = \infty$.*

This example indicates that even if the regularization is arbitrarily small, the hyper-objective before and after regularization can be completely different objectives. Consequently, BLO without LLSC should be treated as a distinct research topic from BLO with LLSC.

### 4.2 Can we Find an LL Optimal Solution?

The goal of finding an LL optimal solution for a given $x \in \mathbb{R}^{d_x}$ is to solve the following problem:

$$\min_{y \in Y^*(x)} f(x, y), \quad Y^*(x) = \arg\min_{y \in \mathcal{Y}} g(x, y). \tag{4}$$

This problem is usually called simple BLO (Beck & Sabach, 2014; Sabach & Shtern, 2017; Kaushik & Yousefian, 2021) since it involves only one variable $y$. However, it is not a "simple" problem as the following theorem shows its intractability for general convex objectives.

**Proposition 4.2.** *Fix an $x$. For any $K \in \mathbb{N}^+$, there exists $d_y \in \mathbb{N}^+$, such that for any $y_0 \in \mathbb{R}^{d_y}$, there exist an 1-Lipschitz linear function $f(x, \cdot)$ and an 1-gradient Lipschitz convex function $g(x, \cdot)$ such that for any first-order algorithm $\mathcal{A}$ which initializes from $y_0 \in \mathcal{Y}$ with $\text{dist}\left(y_0, \arg\min_{y \in Y^*(x)} f(x, y)\right) \leq \sqrt{2}$ and generates a sequence of test points $\{y_k\}_{k=0}^K$ with*

$$y_k \in y_0 + \text{Span}\{\nabla_y f(x, y_0), \nabla_y g(x, y_0), \cdots, \nabla_y f(x, y_{k-1}), \nabla_y g(x, y_{k-1})\}, \quad k \geq 1,$$

*it holds that $|f(x, y_k) - \varphi(x)| \geq 1$.*

The key idea in the proof is to construct the LL function using the worst-case convex zero chain (Nesterov, 2018), such that any first-order algorithm will require a large number of steps to approach the vicinity of the LL solution mapping $Y^*(x)$. The proof is provided in Appendix D, where we also prove a similar lower bound for Lipschitz LL objectives.

### 4.3 Can we Find a UL Stationary Point?

Besides the difficulty in finding an LL optimal solution, the goal of finding a UL stationary point is also challenging. Below, we show that the hyper-objective $\varphi(x)$ can be discontinuous without LLSC. Since continuity is one of the basic assumptions for almost all numerical optimization schemes (Nocedal & Wright, 1999), our hard instance indicates that $\varphi(x)$ may be intrinsically intractable to optimize for BLO without LLSC.

**Example 4.1.** *Consider a BLO instance given by*

$$\min_{x \in \mathbb{R}, y \in Y^*(x)} x^2 + y, \quad Y^*(x) = \arg\min_{y \in [-1,1]} -xy.$$

*The resulting hyper-objective $\varphi(x) = x^2 + \text{sign}(x)$ is discontinuous at $x = 0$.*

In the above example, the discontinuity of $\varphi(x)$ comes from the discontinuity of $Y^*(x) = \text{sign}(x)$. Below, we prove that this statement and its reverse generally holds.

**Proposition 4.3.** *Suppose the solution mapping $Y^*(x)$ is non-empty and compact for any $x \in \mathbb{R}^{d_x}$.*

  a. *If $f(x, y)$ and $Y^*(x)$ are locally Lipschitz, then $\varphi(x)$ is locally Lipschitz.*

b. *Conversely, if $\varphi(x)$ is locally Lipschitz for any locally Lipschitz function $f(x,y)$, then $Y^*(x)$ is locally Lipschitz.*

c. *If $f(x,y)$ is $C_f$-Lipschitz and $Y^*(x)$ is $\kappa$-Lipschitz, then $\varphi(x)$ is $C_\varphi$-Lipschitz with coefficient $C_\varphi = (\kappa + 1)C_f$.*

d. *Conversely, if $\varphi(x)$ is $C_\varphi$-Lipschitz for any $C_f$-Lipschitz function $f(x,y)$, then $Y^*(x)$ is $\kappa$-Lipschitz with coefficient $\kappa = C_\varphi/C_f$.*

Local Lipschitz continuity ensures UL stationary points (Definition 3.8) are well-defined, while global Lipschitz continuity enables uniform complexity bounds for non-asymptotic analysis (as we will use in Section 6.2). According to the above theorem, ensuring the continuity of $Y^*(x)$ is the key to obtaining the desired continuity of $\varphi(x)$. This motivates us to focus on well-behaved LL functions that confer continuity of $Y^*(x)$.

## 5 SUFFICIENT CONDITIONS FOR TRACTABILITY

### 5.1 REGULARITY CONDITIONS FOR CONTINUITY

Since the constructions of the hard instances in the previous section all rely on very *flat* LL functions, our results underscore that *sharpness* of LL functions is essential to ensure the tractability of BLO. This observation inspires us to focus on more restricted function classes that possess sharpness to circumvent the ill-conditioned nature of BLO without LLSC. Below, we introduce two conditions that correspond to different degrees of sharpness.

**Assumption 5.1** (Gradient Dominance). *Suppose $g(x,y)$ is $L$-gradient Lipschitz jointly in $(x,y)$, and there exists $\alpha > 0$ such that for any $x \in \mathbb{R}^{d_x}$, $y \in \mathcal{Y}$ we have $\mathcal{G}_{1/L_g}(y;x) \geq \alpha \mathrm{dist}(y, Y^*(x))$.*

**Assumption 5.2** (Weak Sharp Minimum). *Suppose $g(x,y)$ is $L$-Lipschitz in $x$, and there exists $\alpha > 0$ such that for any $x \in \mathbb{R}^{d_x}$, $y \in \mathcal{Y}$ we have $g(x,y) - g^*(x) \geq 2\alpha \mathrm{dist}(y, Y^*(x))$.*

Both conditions are widely used in convex optimization (Burke & Ferris, 1993; Drusvyatskiy & Lewis, 2018). They are milder conditions than LLSC by allowing $Y^*(x)$ to be non-singleton. Despite being more relaxed, we demonstrate below that either of them can lead to the continuity of $Y^*(x)$ and thus $\varphi(x)$. The continuity of $\varphi(x)$ is crucial for designing algorithms to optimize it.

**Proposition 5.1.** *Under Assumption 5.1 or 5.2, $Y^*(x)$ is $(L/\alpha)$-Lipschitz. Furthermore, if $f(x,y)$ is $C_f$-Lipschitz, then $\varphi(x)$ is $(L/\alpha + 1)C_f$-Lipschtz.*

Therefore, the introduced conditions can avoid discontinuous instances such as Example 4.1. It is worth noting that these conditions fundamentally differ from LLSC, as $\varphi(x)$ can be nonsmooth under these conditions, as exemplified below. The potential nonsmoothness of $\varphi(x)$ further justifies the rationality of using Goldstein stationarity in Definition 3.8.

**Example 5.1.** *Let $f(x,y) = xy, g(x,y) = 0$ and $\mathcal{Y} = [-1, 1]$. We obtain a BLO instance satisfying both Assumption 5.1 and 5.2. But the resulting $\varphi(x) = -|x|$ is nonsmooth and nonconvex.*

### 5.2 HOW TO VERIFY THE CONDITIONS?

One may wonder how to verify the introduced conditions in applications. It is non-trivial as the value of $\mathrm{dist}(y, Y^*(x))$ is unknown. An easy case is Assumption 5.1 with $\mathcal{Y} = \mathbb{R}^{d_y}$, which reduces to the Polyak-Łojasiewicz condition (Polyak, 1963): $\|\nabla_y g(x,y)\|^2 \geq 2\alpha(g(x,y) - g^*(x))$ by Theorem 2 in Karimi et al. (2016). This inequality allows us to identify the following examples that fall into Assumption 5.1. Firstly, we can show that Assumption 5.1 strictly covers the LLSC condition.

**Example 5.2.** *If $g$ is $L$-gradient Lipschitz and $\alpha$-strongly convex, then it satisfies Assumption 5.1.*

Secondly, the following example that both AID and ITD fail to optimize satisfies Assumption 5.1.

**Example 5.3.** *Consider the hard BLO instance proposed by Liu et al. (2020):*

$$\min_{x \in \mathbb{R}, y \in Y^*(x)} (x - y_{[2]})^2 + (y_{[1]} - 1)^2, \quad Y^*(x) = \arg\min_{y \in \mathbb{R}^2} y_{[1]}^2 - 2xy_{[1]}.$$

*The LL function satisfies Assumption 5.1 with $L = 1$ and $\alpha = 1/4$.*

Thirdly, the BLO with least squares loss studied by Bishop et al. (2020) also satisfies Assumption 5.1. We leave more details of this model and its application in adversarial training in Section 7.1.

**Example 5.4.** *Consider the BLO with least squares loss:*

$$\min_{x\in\mathbb{R}^{d_x}, y\in Y^*(x)} \frac{1}{2n}\|Ax-y\|^2, \quad Y^*(x) = \arg\min_{y\in\mathbb{R}^{d_y}} \frac{1}{2n}\|Ax-y\|_M^2 + \frac{\lambda}{2n}\|y-b\|_M^2,$$

*where $A\in\mathbb{R}^{n\times d_x}$, $b\in\mathbb{R}^n$ represents the features and labels of the $n$ samples in the dataset, $\lambda>0$ and $M$ is a positive semi-definite matrix that induces the norm $\|z\|_M = \sqrt{z^\top M z}$. The LL function satisfies Assumption 5.1 with $L=(\lambda+1)\sigma_{\max}(M)$ and $\alpha=(\lambda+1)\sigma_{\min}^+(M)$.*

## 6 THE PROPOSED METHODS

In this section, we propose novel polynomial time algorithms for BLO under Assumption 5.1 and 5.2. In Section 6.1, we borrow ideas from switching gradient methods to overcome the difficulty of multiple LL minima. In Section 6.2 we propose a method motivated by zeroth-order optimization that can provably converge to a UL stationary point.

### 6.1 FINDING AN LL OPTIMAL SOLUTION VIA SWITCHING GRADIENT METHOD

---

**Algorithm 1** SGM $(x, y_0, K_0, K, \tau, \theta)$

---

1: $\mathcal{I} = \emptyset$, $\hat{y}_0 = y_0$
2: **for** $k = 0, 1, \cdots, K_0 - 1$
3:    $\hat{y}_{k+1} = \mathcal{P}_\mathcal{Y}(\hat{y}_k - \tau\partial_y g(x, \hat{y}_k))$
4: **end for**
5: $\hat{g}^*(x) = g(x, \hat{y}_{K_0})$
6: **for** $k = 0, 1, \cdots, K - 1$
7:    **if** $g(x, y_k) - \hat{g}^*(x) \le 2\theta$
8:       $y_{k+1} = \mathcal{P}_\mathcal{Y}(y_k - \tau\partial_y f(x, y_k))$
9:       $\mathcal{I} = \mathcal{I} \cup \{k\}$
10:   **else**
11:      $y_{k+1} = \mathcal{P}_\mathcal{Y}(y_k - \tau\partial_y g(x, y_k))$
12: **end for**
13: $y_{\text{out}} = \frac{1}{|\mathcal{I}|} \sum_{k\in\mathcal{I}} y_k$
14: **return** $y_{\text{out}}$

---

In Equation 4, the LL constraint $y \in Y^*(x)$ is equivalent to an inequality constraint $g(x, y) \le g^*(x)$. Based on this observation, we generalize Polyak's Switching Gradient Method (Polyak, 1967) for functional constrained problems to Algorithm 1 when the following assumptions hold.

**Assumption 6.1.** *Suppose that*

    *a. both $f(x, y)$ and $g(x, y)$ are convex in $y$;*

    *b. $\mathcal{Y}$ is compact with diameter $R$;*

    *c. $f(x, y)$ is $C_f$-Lipschitz on $\mathbb{R}^{d_x} \times \mathcal{Y}$;*

    *d. $g(x, \cdot)$ is $C_g$-Lipschitz on $\mathcal{Y}$ for any $x \in \mathbb{R}^{d_x}$;*

    *e. either Assumption 5.1 or 5.2 holds for $g(x, y)$.*

Under the above assumptions, we can prove the following result.

**Theorem 6.1.** *Fix an $x$. Under Assumption 6.1, Algorithm 1 with appropriate parameters can ouput a point $y_{\text{out}}$ satisfying $|f(x, y_{\text{out}}) - \varphi(x)| \le \zeta$ and $g(x, y_{\text{out}}) - g^*(x) \le \zeta$ with $\mathcal{O}(\text{poly}(1/\zeta))$ first-order oracle calls from $g$.*

The corresponding proof and specific parameters of the algorithm can be found Appendix G.

---

**Algorithm 2** IGFM $(x_0, y_0, \eta, T, \delta, K_0, K, \tau, \theta)$

---

1: **Require:** Sub-routine $\mathcal{A}$ can estimate $\tilde{\varphi}(x) \approx \varphi(x)$ for any $x \in \mathbb{R}^{d_x}$
2: **for** $t = 0, 1, \cdots, T - 1$
3:     Sample $u_t \in \mathbb{R}^{d_x}$ uniformly from the unit sphere in $\mathbb{R}^{d_x}$.
4:     Estimate $\tilde{\varphi}(x_t + \delta u_t)$ and $\tilde{\varphi}(x_t - \delta u_t)$ by sub-routine $\mathcal{A}$.
5:     $\hat{\nabla}_t = \frac{d_x}{2\delta} \left( \tilde{\varphi}(x_t + \delta u_t) - \tilde{\varphi}(x_t - \delta u_t) \right) u_t$
6:     $x_{t+1} = x_t - \eta \hat{\nabla}_t$
7: **end for**
8: **return** $x_{\text{out}}$ uniformly chosen from $\{x_t\}_{t=0}^{T-1}$

---

## 6.2 FINDING A UL STATIONARY POINT VIA ZEROTH-ORDER METHOD

Without LLSC, the hyper-gradient $\nabla\varphi(x)$ may not have an explicit form as Equation 3. To tackle this challenge, we propose the Inexact Gradient-Free Method (IGFM) in Algorithm 2. The algorithm is motivated by recent advances in nonsmooth nonconvex zeroth-order optimization (Lin et al., 2022). Our zeroth-order oracle $\tilde{\varphi}(x) \approx \varphi(x)$ is "inexact" since it is an approximation from a sub-routine $\mathcal{A}$. Below, we show that when $\mathcal{A}$ can guarantee sufficient approximation precision, the IGFM can provably find a Goldstein stationary point of a Lipschitz hyper-objective function $\varphi(x)$.

**Assumption 6.2.** *Suppose that*

    *a.  $\varphi(x)$ is $C_\varphi$-Lipschitz.*

    *b.  $\mathcal{A}$ ensures $|\tilde{\varphi}(x) - \varphi(x)| \leq \mathcal{O}(\delta\varepsilon^2/(d_x C_\varphi))$ for any $x \in \mathbb{R}^{d_x}$.*

**Theorem 6.2.** *Given any $\varepsilon \lesssim C_f$. Let $\Delta = \varphi(x_0) - \varphi^*$. Under Assumption 6.2, set*

$$T = \mathcal{O}\left( d_x^{3/2} \left( \frac{C_\varphi^4}{\varepsilon^4} + \frac{\Delta C_\varphi^3}{\delta\varepsilon^4} \right) \right), \ \eta = \Theta\left( \sqrt{\frac{\delta(\Delta + \delta C_\varphi)}{d_x^{3/2} C_\varphi^3 T}} \right). \tag{5}$$

*Then Algorithm 2 can output a point $x_{\text{out}}$ that satisfies $\mathbb{E} \min \{\|s\| : s \in \partial_\delta \varphi(x_{\text{out}})\} \leq \varepsilon$.*

Now it remains to verify Assumption 6.2: Assumption 6.2a can be satisfied by Proposition 4.3, while Assumption 6.2b can be satisfied by Theorem 6.1. Therefore, we have the following result.

**Corollary 6.1.** *Suppose Assumption 6.1 holds. Set $\mathcal{A}$ as Algorithm 1. Then Algorithm 2 with appropriate parameters can output a $(\delta, \epsilon)$-Goldstein stationary point of $\varphi(x)$ in expectation within $\mathcal{O}(\mathrm{poly}(d_x, 1/\varepsilon, 1/\delta))$ zeroth-order and first-order oracle calls from $f$ and $g$.*

To the best of our knowledge, it is the first theoretical analysis that shows the non-asymptotic convergence to a UL stationary point for BLO without LLSC.

## 7 NUMERICAL EXPERIMENTS

Table 2: MSE (mean $\pm$ std) achieved by different algorithms on the "abalone" dataset in adversarial training.

| Method | MSE |
|--------|-----|
| AID | $1.781 \pm 0.418$ |
| ITD | $0.982 \pm 0.015$ |
| BGS | $0.995 \pm 0.259$ |
| BDA | $0.976 \pm 0.014$ |
| BOME | $0.999 \pm 0.140$ |
| IA-GM | $0.992 \pm 0.013$ |
| IGFM (Ours) | $\mathbf{0.936 \pm 0.015}$ |

Table 3: Test accuracy (%) achieved by different algorithms on the MNIST dataset under different corruption rates $p$ in hyperparameter tuning.

| Method | $p = 0.5$ | $p = 0.3$ | $p = 0.1$ |
|--------|-----------|-----------|-----------|
| AID | 75.8 | 87.5 | 91.3 |
| ITD | 75.8 | 87.5 | 91.3 |
| BGS | 75.8 | 87.5 | 91.3 |
| BDA | 81.2 | 89.3 | 91.5 |
| BOME | 86.7 | 88.9 | 89.3 |
| IA-GM | 86.9 | 90.3 | 90.5 |
| IGFM (Ours) | **88.4** | **91.0** | **91.8** |

In this section, we compare IGFM with different baselines, including AID with conjugate gradient (Maclaurin et al., 2015), ITD (Ji et al., 2021), BGS (Arbel & Mairal, 2022), BDA (Liu et al.,

2020), BOME (Liu et al., 2022), and IA-GM (Liu et al., 2021) in the following two different applications of BLO without LLSC.

## 7.1 ADVERSARIAL TRAINING

Brückner & Scheffer (2011) proposed modeling adversarial training via BLO. In this model, the learner aims at finding the optimal parameter $x$, subject to data $y$ being modified by an adversarial data provider. Like Bishop et al. (2020); Wang et al. (2021; 2022a), we use least squares loss for both $f$ and $g$ as Example 5.4. In the LL loss, we use a diagonal matrix $M$ to assign different weights to each sample, and a ridge term $\|y - b\|_M^2$ to penalize the data provider when manipulating the original labels $b$. We set half the diagonal elements of $M$ evenly in $[\sigma_{\min}^+, \sigma_{\max}]$ and the rest zero. We let $\lambda = 1$, $\sigma_{\max} = 1$ and $\sigma_{\min}^+ = 10^{-9}$. For BDA, we choose $s_u = s_l = 1$, $\alpha_k = \mu/(k+1)$ and tune $\mu$ from $\{0.1, 0.5, 0.9\}$ as Liu et al. (2020). For BOME, we choose the default option for $\phi_k$ and $\eta$ from $\{0.9, 0.5, 0.1\}$ as Liu et al. (2022). For IGFM, we choose $\delta = 10^{-3}$ and tune $\theta$ from $\{10^{-1}, 10^{-2}, 10^{-3}\}$. For all algorithms, we tune the learning rates in $\{10^2, 10^1, 10^0, 10^{-1}, 10^{-2}, 10^{-3}, 10^{-4}, 10^{-5}\}$. We run all the algorithms for 500 UL iterations, with 10 LL iterations per UL iteration. Table 2 compares the mean squared error (MSE), measured by the value of $\varphi(x)$, achieved by the algorithms on the "abalone" dataset from LIBSVM (Chang & Lin, 2011). AID has poor performance because it requires taking the inverse of $\nabla_{yy}^2 g(x, y)$, which is ill-conditioned in this experiment. Among all the algorithms, the IGFM achieves the lowest mean value of MSE, and its variance is also maintained at a relatively low level.

## 7.2 HYPERPARAMETER TUNING

We consider tuning the optimal $\ell_2$ regularization for logistic regression to avoid overfitting a noisy training set $\mathcal{D}^{\mathrm{tr}}$, based on the performance on a clean validation set $\mathcal{D}^{\mathrm{val}}$. We let the UL variable $x$ be the log-transformed regularization coefficient to avoid the constraint $x \geq 0$ (Pedregosa, 2016; Bertrand et al., 2020), and the LL variable $y$ be the weight of the model. The problem can be formulated as BLO with:

$$f(x, y) = \frac{1}{|\mathcal{D}^{\mathrm{val}}|} \sum_{(a_i, b_i) \in \mathcal{D}^{\mathrm{val}}} \ell(\langle a_i, y \rangle, b_i),$$

$$g(x, y) = \frac{1}{|\mathcal{D}^{\mathrm{tr}}|} \sum_{(a_i, b_i) \in \mathcal{D}^{\mathrm{tr}}} \ell(\langle a_i, y \rangle, b_i) + \exp(x)\|y\|^2,$$

where $(a_i, b_i)$ is the $i$-th feature-label pair in the dataset, and $\ell(\cdot, \cdot)$ is the cross-entropy loss. We use the MNIST dataset (LeCun, 1998) in this experiment. We use 40,000 images for $\mathcal{D}^{\mathrm{tr}}$ and 20,000 images for $\mathcal{D}^{\mathrm{val}}$. We corrupt $\mathcal{D}^{\mathrm{tr}}$ by assigning random labels with probability $p$ (Liu et al., 2022). We follow the same hyperparameter selection strategy as Section 7.1, and run all the algorithms with 100 UL iterations. Table 3 reports the accuracy evaluated on the testing set with 10,000 images under different levels of $p$. It can be seen that IGFM achieves the highest accuracy among all algorithms. Note that AID / ITD / BGS have similar performances since AID and ITD are proven to be consistent under LLSC (Ji et al., 2021) and BGS is a combination of them.

## 8 CONCLUSIONS AND DISCUSSIONS

This paper gives a comprehensive study of BLO without the typical LLSC assumption. We provide hardness results to show the intractability of this problem and introduce several key regularity conditions that can confer tractability. Novel algorithms with non-asymptotic convergence are proposed as well. Experiments on real-world datasets support our theoretical investigations.

Although this paper focuses primarily on the theoretical level, we expect our theory can shed light on efficient algorithm design for BLO applications in practice. We also hope our work can be a good starting point for non-asymptotic analysis for more challenging BLO problems, such as BLO with nonconvex LL functions or BLO with intertwined inequality constraints $h(x, y) \leq 0$.

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
