# A    LIMITATIONS OF VALUE-FUNCTION APPROACH

In contrast to the hyper-objective approach adopted in this paper that pursues a UL stationary point such that $\|\nabla\varphi(x)\| \leq \varepsilon$, existing non-asymptotic analysis (Liu et al., 2022; Sow et al., 2022) for BLO without LLSC relies on following value-function reformulation for Problem 1:

$$\min_{x \in \mathbb{R}^{d_x}, y \in \mathbb{R}^{d_y}} f(x, y) \quad \text{s.t.} \quad g(x, y) - g^*(x) \leq 0. \tag{6}$$

These value-function approaches show convergence to the following KKT points.

**Definition A.1** (KKT point). *Suppose that $g^*(x)$ is Clarke subdifferentiable. We call $(x, y)$ an $\varepsilon$-KKT point of Problem 6 if there exists a scalar $\lambda \geq 0$ such that we have*

    *a. (Stationary in x) $\|\nabla_x f(x, y) + \lambda(\nabla_x g(x, y) - \partial g^*(x))\| \leq \varepsilon$.*

    *b. (Stationary in y) $\|\nabla_y f(x, y) + \lambda\nabla_y g(x, y)\| \leq \varepsilon$.*

    *c. (Feasibility) $g(x, y) - g^*(x) \leq \varepsilon$.*

    *d. (Complementary Slackness) $|\lambda(g(x, y) - g^*(x))| \leq \varepsilon$.*

*We call $(x, y)$ a KKT point if $\varepsilon = 0$.*

**Remark A.1.** *In Definition A.1 we assume that $g^*(x)$ is Clarke differentiable. It can be easily satisfied under some mild conditions. For instance, when $g(x, y)$ is L-gradient Lipschitz, $g^*(x)$ is provably L-weakly concave, and thus Clarke differentiable (Rockafellar & Wets, 2009). In the unconstrained case that $\mathcal{Y} = \mathbb{R}^{d_y}$, under LLSC or more generally under Assumption 5.1, $g^*(x)$ is provably differentiable (Nouiehed et al., 2019) and the Clarke subdifferential $\partial g^*(x)$ reduces to the classical gradient $\nabla g^*(x)$.*

Unfortunately, classical constraint qualifications provably fail for the value-function-based reformulation (Ye & Zhu, 1995). For this reason, we can easily construct a BLO instance whose KKT points do not contain the optimal solution even under LLSC.

**Example A.1.** *Consider a BLO instance given by:*

$$\min_{x \in \mathbb{R}, y \in \mathbb{R}} -xy, \quad \text{s.t.} \ (x + y - 2)^2 \leq 0,$$

*where the lower-level function is strongly convex in y. For this example:*

    *a. The stationary point of $\varphi(x)$ is exactly the global solution $x^*$.*

    *b. However, the KKT points by Definition A.1 do not include any solution to this problem.*

*Proof.* We know that the lower-level constraint is $y = 2 - x$, so the problem is equivalent to $\min_{x \in \mathbb{R}} x^2 - 2x$ with the unique solution $(x^*, y^*) = (1, 1)$. However, if we rewrite the problem by:

$$\min_{x \in \mathbb{R}, y \in \mathbb{R}} -xy, \quad \text{s.t.} \ (x + y - 2)^2 \leq 0.$$

The KKT condition is

$$\begin{cases} y - 2\lambda(x + y - 2) = 0; \\ x - 2\lambda(x + y - 2) = 0; \\ \lambda(x + y - 2)^2 = 0; \\ (x + y - 2)^2 \leq 0, \ \lambda \geq 0. \end{cases}$$

When $\lambda > 0$ there is no $(x, y)$ that satisfies the KKT condition. When $\lambda = 0$, the KKT condition is only satisfied by $(x, y) = (0, 0)$, but it is not the solution to this problem. □

One may argue that when relaxing the goal into finding an $\varepsilon$-KKT point, Slater's constraint qualification can be satisfied since we allow the constraint $g(x, y) - g^*(x) \leq 0$ to be violated slightly. However, we give a concrete example indicating that an $\varepsilon$-KKT point may be far away from the solution set, even when the hyper-objective $\varphi(x)$ is strongly convex.

**Example A.2.** *Given $0 < \varepsilon \leq 1$. Suppose $\varphi(x)$ is $\mu$-strongly convex with a unique solution $x^*$.*

    *a. Whenever a given point $x$ satisfies $\|\nabla\varphi(x)\| \leq \varepsilon$, we have $\|x - x^*\| \leq \varepsilon/\mu$.*

    *b. However, there exists a BLO instance with a convex lower-level function such that the resulting $\varphi(x)$ is strongly convex, but there is an infinite number of $2\varepsilon$-stationary points $(x, y)$ by Definition A.1 such that $\|x - x^*\| = 1$.*

*Proof.* Below we prove the two parts separately.

**a.** Strong convexity ensures that $\mu\|x - x^*\| \leq \|\nabla\varphi(x)\|$.

**b.** Consider the bilevel problem given by:

$$\min_{x \in \mathbb{R}, y \in \mathbb{R}} x^2 - 2\varepsilon xy, \quad \text{s.t. } y \in \arg\min_{y \in \mathbb{R}} \varepsilon^3 y^2,$$

where the lower-level problem is convex in $y$ and the global solution is $x^* = 0$. It can be verified that $(x, y) = (1, \varepsilon^{-1})$ is an $\varepsilon$-KKT point with any multiplier satisfying $0 < \lambda \leq 1$ by

$$\begin{cases} g(x, y) - g^*(x) = \varepsilon^3 y^2 = \varepsilon; \\ |\nabla_x f(x, y) + \lambda(\nabla_x g(x, y) - \nabla g^*(x))| = 2(x - \varepsilon y) = 0; \\ |\nabla_y f(x, y) + \lambda \nabla_y g(x, y)| = 2(\varepsilon x - \lambda \varepsilon^3 y) \leq 2\varepsilon. \end{cases}$$

But we know that $\|x - x^*\| = 1$. $\qquad\square$

## B   PROOFS IN SECTION 3

**Proposition 3.1.** *Suppose $f(x, \cdot)$ and $g(x, \cdot)$ are convex, and $\varphi(x)$ is locally Lipschitz. Then for any locally optimistic optimal solution $(x^*, y^*)$, we have $\partial\varphi(x^*) = 0$, $f(x^*, y^*) = \varphi(x^*)$ and $g(x^*, y^*) = g^*(x^*)$.*

*Proof.* By the definition that $y^* \in Y^*(x)$, we know that $g(x^*, y^*) = g^*(x^*)$. When $g(x, \cdot)$ is convex, we know that $Y^*(x)$ is also a convex set for any given $x$. Then the problem $\min_{y \in Y^*(x)} f(x, y)$ is a convex problem with respect to $y$, where a local minimum is also a global minimum. This indicates that $\varphi(x^*) = f(x^*, y^*)$. Finally, the first-order necessary optimality condition for a local minimum of $\varphi(x)$ implies that $\partial\varphi(x^*) = 0$ (Theorem 8.4 by Clason (2017)).

$\qquad\square$

## C   PROOFS IN SECTION 4.1

**Proposition 4.1.** *Given a pivot $\hat{y}$, there exists a BLO instance, where both $f(x, y)$ and $g(x, y)$ are convex in $y$, and the resulting hyper-objective $\varphi(x)$ is a quadratic function, but for any $\lambda > 0$ the regularized hyper-objective*

$$\varphi_\lambda(x) = \min_{y \in Y^*_\lambda(x)} f(x, y), \quad Y^*_\lambda(x) = \arg\min_{y \in \mathcal{Y}} g(x, y) + \lambda\|y - \hat{y}\|^2$$

*is a linear function with $|\inf_{x \in \mathbb{R}^{d_x}} \varphi_\lambda(x) - \inf_{x \in \mathbb{R}^{d_x}} \varphi(x)| = \infty$.*

*Proof.* We distinguish two different cases by whether we have $\hat{y}_{[1]} = 0$.

When $\hat{y}_{[1]} \neq 0$, we consider the problem given by

$$\min_{x \in \mathbb{R}, y \in Y^*(x)} y_{[1]}^2 - 2xy_{[1]}, \quad Y^*(x) = \arg\min_{y \in \mathbb{R}^2} (y_{[2]} - \hat{y}_{[2]})^2.$$

After adding regularization, we have $Y^*_\lambda(x) = \{\hat{y}\}$ and $\varphi_\lambda(x) = \hat{y}_{[1]}^2 - 2x\hat{y}_{[1]}$.

When $\hat{y}_{[1]} = 0$, we instead consider the problem given by

$$\min_{x \in \mathbb{R}, y \in Y^*(x)} (y_{[1]} + 1)^2 - 2x(y_{[1]} + 1), \quad Y^*(x) = \arg\min_{y \in \mathbb{R}^2} (y_{[2]} - \hat{y}_{[2]})^2.$$

And after adding regularization we have $Y_\lambda^*(x) = \{0\}$ and $\varphi_\lambda(x) = 1 - 2x$.

However, for both the two cases the original hyper-objective is the quadratic function $\varphi(x) = -x^2$.

$\square$

## D    PROOFS IN SECTION 4.2

Our lower bound is based on the following first-order zero-chain (Nesterov, 2018).

**Definition D.1** (Zero-Chain). *We call function $h(z) : \mathbb{R}^q \to \mathbb{R}$ a first-order zero-chain if for any sequence $\{z_k\}_{k \geq 1}$ satisfying*

$$z_i \in \text{Span}\{\partial h(z_0), \cdots, \partial h(z_{i-1})\}, \quad i \geq 1; \quad z_0 = 0 \tag{7}$$

*it holds that $z_{i,[j]} = 0$, $i + 1 \leq j \leq q$.*

We remark that in the construction of a zero chain, we can always assume $z_0 = 0$ without loss of generality. Otherwise, we can translate the function to $h(z - z_0)$. Below, we introduce the convex zero-chain from Nesterov (2018), Section 2.1.2.

Since the subgradients may contain more than one element, we also say $h(z)$ is zero-chain whenever there exists some adversarial subgradient oracle. This would also provide a valid lower bound (Nesterov, 2018).

**Definition D.2** (Gradient Lipschitz Worse-Case Zero-Chain). *Consider the family of functions:*

$$h_q(z) = \frac{1}{8}(z_{[1]} - 1)^2 + \frac{1}{8}\sum_{j=1}^{q-1}\left(z_{[j+1]} - z_{[j]}\right)^2.$$

*The following properties hold for any $h_q(z)$ with $q \in \mathbb{N}^+$:*

> *a. It is a first-order zero-chain.*
>
> *b. It has a unique minimizer $z^* = \mathbf{1}$.*
>
> *c. It is $1$-gradient Lipschitz.*

In bilevel problems, it is crucial to find a point $y$ that is close to $Y^*(x)$, instead of just achieving a small optimality gap $g(x, y) - g^*(x)$. However, it is difficult for any first-order algorithms to "locate" the minimizers of the function class in Definition D.2.

**Proposition 4.2.** *Fix an $x$. For any $K \in \mathbb{N}^+$, there exists $d_y \in \mathbb{N}^+$, such that for any $y_0 \in \mathbb{R}^{d_y}$, there exists a pair of functions $f(x, \cdot), g(x, \cdot)$ that are both convex and $1$-gradient Lipschitz , for any first-order algorithm $\mathcal{A}$ which initializes from $y_0 \in \mathcal{Y}$ with $\text{dist}\,(y_0, y^*(x)) \leq 1$ and generates a sequence of test points $\{y_k\}_{k=0}^K$ with*

$$y_k \in y_0 + \text{Span}\{\nabla_y f(x, y_0), \nabla_y g(x, y_0), \cdots, \nabla_y f(x, y_{k-1}), \nabla_y g(x, y_{k-1})\}, \quad k \geq 1,$$

*it holds that $|f(x, y_k) - \varphi(x)| \geq 1/4$, where $y^*(x)$ is the unique solution to $\min_{y \in Y^*(x)} f(x, y)$.*

*Proof.* Without loss of generality, we assume $y_0 = 0$. Let $d_y = q = 2K$, $\sigma = 1/\sqrt{q}$ and

$$f(x, y) = \frac{1}{2}\sum_{j=K+1}^{q} y_{[j]}^2, \quad g(x, y) = \sigma^2 h_q(y/\sigma),$$

where $h_q(y)$ follows Definition D.2. It is clear from the construction that both $f(x, \cdot), g(x, \cdot)$ are convex and $1$-gradient Lipschitz. Moreover, both of them are zero-chains. Then the property of zero-chain leads to

$$y_{k,[j]} = 0, \quad \forall k + 1 \leq j \leq q, \quad 0 \leq k \leq K.$$

Therefore $f(x, y_k)$ remains zero for all $0 \leq k \leq K$.

However, we know that $Y^*(x) = \{\sigma\mathbf{1}\}$. Therefore,

$$\varphi(x) = \frac{1}{2} \sum_{j=K+1}^{q} \sigma^2 = \frac{K\sigma^2}{2} = \frac{1}{4},$$

which indicates that any first-order algorithm $\mathcal{A}$ has a constant sub-optimality gap. $\square$

Next, we prove similar a lower bound also holds for Lipschitz nonsmooth convex lower-level functions, using the following function class, which appears in Nesterov (2018), Section 3.2.1.

**Definition D.3** (Lipschitz Zero-Chain). *Consider the family of functions:*

$$h_q(z) = \frac{\sqrt{q}}{2 + \sqrt{q}} \max_{1 \le j \le q} z_{[j]} + \frac{1}{2\left(2 + \sqrt{q}\right)} \|z\|^2,$$

*The following properties hold for any $h_q(z)$ with $q \in \mathbb{N}^+$:*

a. *It is a first-order zero-chain.*

b. *It has a unique minimizer $z^* = -\mathbf{1}/\sqrt{q}$.*

c. *It is 1-Lipschitz in the unit Euclidean ball $\mathbb{B}(z^*) \triangleq \{z : \|z - z^*\| \le 1\}$.*

Analogous to Proposition 4.2, we can show the following result.

**Proposition D.1.** *Fix an $x$. For any $K \in \mathbb{N}^+$, there exists $d_y \in \mathbb{N}^+$, such that for any $y_0 \in \mathbb{R}^{d_y}$, there exist there exists a* *pair of functions $f(x, \cdot), g(x, \cdot)$ that are both convex and 1-Lipschitz* *on $\mathbb{B}(y^*(x))$, such that for any first-order algorithm $\mathcal{A}$ which initializes from $y_0 \in \mathbb{B}(y^*(x))$, and generates a sequence of test points $\{y_k\}_{k=0}^{K}$ with*

$$y_k \in y_0 + \mathrm{Span}\{\partial_y f(x, y_0), \partial_y g(x, y_0), \cdots, \partial_y f(x, y_{k-1}), \partial_y g(x, y_{k-1})\}, \quad k \ge 1,$$

*there exists some subgradients sequence $\{\partial_y f(x, y_0), \partial_y g(x, y_0), \cdots, \partial_y g(x, y_{k-1})\}$ to make $|f(x, y_k) - \varphi(x)| \ge 1/4$ for all $k$, where $y^*(x)$ is the unique solution to $\min_{y \in Y^*(x)} f(x, y)$.*

*Proof.* Without loss of generality, we assume $y_0 = 0$. Let $d_y = q = 2K$ and

$$f(x, y) = \sum_{j=K+1}^{q} \psi(y_{[j]}), \quad g(x, y) = h_q(y),$$

where $h_q(y)$ follows Definition D.3 and $\psi(y)$ the Huber function defined by

$$\psi(y) = \begin{cases} \beta y - \frac{1}{2} y^2, & y \ge \beta; \\ \frac{1}{2} y^2, & -\beta < y < \beta; \\ -\beta y + \frac{1}{2} y^2, & y \le -\beta. \end{cases}$$

Since $|\psi'(y)| \le \beta$, we know $f(x, \cdot)$ is $(\sqrt{q}\beta)$-Lipschitz since

$$\left| \sum_{j=K+1}^{q} \psi(y_{[j]}) - \sum_{j=K+1}^{q} \psi(y'_{[j]}) \right|$$

$$\le \sum_{j=K+1}^{q} \left| \psi(y_{[j]}) - \psi(y'_{[j]}) \right|$$

$$\le \beta \sum_{j=K+1}^{q} \left| y_{[j]} - y'_{[j]} \right|$$

$$\le \beta\sqrt{q}\|y - y'\|$$

Let $\beta = 1/\sqrt{q}$ then $f(x, \cdot)$ is 1-Lipschitz. And $g(x, \cdot)$ is 1-Lipschitz on $\mathbb{B}(y^*(x))$.

Note that $f$ always returns a zero subgradient at the origin, while $g$ is a zero-chain. We have

$$y_{k,[j]} = 0, \quad \forall k + 1 \le j \le q, \quad 0 \le k \le K.$$

Therefore $f(x, y_k)$ remains zero for all $0 \le k \le K$.

However, we know that $Y^*(x) = \{-\mathbf{1}/\sqrt{q}\}$. So it can be calculated that

$$\varphi(x) = \sum_{j=K+1}^{q} \psi(-1/\sqrt{q}) = -\frac{K}{2q} = -\frac{1}{4},$$

which indicates that any first-order algorithm $\mathcal{A}$ has a constant sub-optimality gap. $\square$

We remark that projection onto the ball centered at the origin $\mathbb{B}(0)$ will not produce additional nonzero entries. Therefore, the possible projection operation in the algorithm will not distort the zero-chain structure.

## E  PROOFS IN SECTION 4.3

**Proposition 4.3.** *Suppose the solution mapping $Y^*(x)$ is non-empty and compact for any $x \in \mathbb{R}^{d_x}$.*

  a. *If $f(x, y)$ and $Y^*(x)$ are locally Lipschitz , then $\varphi(x)$ is locally Lipschitz.*

  b. *Conversely, if $\varphi(x)$ is locally Lipschitz for any locally Lipschitz function $f(x, y)$, then $Y^*(x)$ is locally Lipschitz.*

  c. *If $f(x, y)$ is $C_f$-Lipschitz and $Y^*(x)$ is $\kappa$-Lipschitz, then $\varphi(x)$ is $C_\varphi$-Lipschitz with coefficient $C_\varphi = (\kappa + 1)C_f$.*

  d. *Conversely, if $\varphi(x)$ is $C_\varphi$-Lipschitz for any $C_f$-Lipschitz function $f(x, y)$ , then $Y^*(x)$ is $\kappa$-Lipschitz with coefficient $\kappa = C_\varphi/C_f$.*

*Proof.* Note that we can replace sup and inf with max and min in Definition 3.3 due to the compactness of $Y^*(x)$. Below we prove each part of the proposition, separately.

**a.** Since $Y^*(x_1), Y^*(x_2)$ are nonempty compact sets, we can pick

$$y_1 \in \arg\min_{y \in Y^*(x_1)} f(x_1, y), \quad y_2 \in \arg\min_{y \in Y^*(x_2)} f(x_2, y).$$

Then the Lipschitz continuity of $Y^*(x)$ implies there exist $y_1' \in Y^*(x_1)$ and $y_2' \in Y^*(x_2)$ such that

$$\varphi(x_1) - \varphi(x_2) \le f(x_1, y_1') - f(x_2, y_2) \le C_f \left( \|x_1 - x_2\| + \|y_2 - y_1'\| \right) \le (\kappa + 1)C_f \|x_1 - x_2\|,$$
$$\varphi(x_2) - \varphi(x_1) \le f(x_2, y_2') - f(x_1, y_1) \le C_f \left( \|x_1 - x_2\| + \|y_1 - y_2'\| \right) \le (\kappa + 1)C_f \|x_1 - x_2\|,$$

This establishes the Lipschitz continuity of $\varphi$.

**b.** It suffices to bound the following term for any $x_1, x_2$:

$$\max \left\{ \underbrace{\max_{y_2 \in Y^*(x_2)} \min_{y_1 \in Y^*(x_1)} \|y_1 - y_2\|}_{(I)}, \quad \underbrace{\max_{y_1 \in Y^*(x_1)} \min_{y_2 \in Y^*(x_2)} \|y_1 - y_2\|}_{(II)} \right\}. \tag{8}$$

Without loss of generality, we assume $C_f = 1$, otherwise we can scale $f(x, y)$ by $C_f$ to prove the result. We let $f(x, y) = -\min_{y_1 \in Y^*(x_1)} \|y - y_1\|$, then

$$(I) = \varphi(x_1) - \varphi(x_2) \le C_\varphi \|x_1 - x_2\|.$$

Next, we let $f(x, y) = \max_{y_1 \in Y^*(x_1)} \|y - y_1\|$, then

$$(II) \le \varphi(x_2) - \varphi(x_1) \le C_\varphi \|x_1 - x_2\|.$$

Together, recalling the definition of (I) and (II) in Equation 8, we know that

$$\text{dist}(Y^*(x_1), Y^*(x_2)) \leq C_\varphi \|x_1 - x_2\|, \quad \forall x_1, x_2 \in \mathbb{R}^d.$$

Proposition 4.3c and Proposition 4.3d replace the global Lipschitz continuity with local Lipschitz continuity. The proofs are similar, with additional care for the local argument.

**c.** We use $\mathcal{N}_\delta(\cdot)$ to denote the open neighbourhood ball with radius $\delta$. For a vector $z$, we define $\mathcal{N}_\delta(z) \triangleq \{z' : \|z' - z\| < \delta\}$. For a set $S$, we define $\mathcal{N}_\delta(S) \triangleq \{z' : \text{dist}(z', S) < \delta\}$. For a given $x_1 \in \mathbb{R}^d$ and any $y \in Y^*(x_1)$, the local Lipschitz continuity of $f(\cdot, \cdot)$ implies that there exists $\delta_y > 0$ and $L_y > 0$ such that $f(\cdot, \cdot)$ is $L_y$-Lipschitz in $\mathcal{N}_{\delta_y}(x_1) \times \mathcal{N}_{\delta_y}(y)$. Note that the set $S \triangleq \cup_y \left\{ \mathcal{N}_{\delta_y}(x_1) \times \mathcal{N}_{\delta_y}(y) \right\}$ forms an open cover of the set $x_1 \times Y^*(x_1)$. The compactness of set $Y^*(x_1)$ guarantees the existence of a finite subcover $\cup_{k=0}^n \left\{ \mathcal{N}_{\delta_{y_k}}(x_1) \times \mathcal{N}_{\delta_{y_k}}(y_k) \right\}$. Therefore, we can conclude that there exists $\delta_1 > 0$, such that $f(\cdot, \cdot)$ is $L_1$-Lipschitz in the neighborhood $\mathcal{N}_{\delta_1}(x_1) \times \mathcal{N}_{\delta_1}(Y^*(x_1))$, where $L_1 = \max_k L_{y_k}$.

Next, the local Lipschitz continuity of $Y^*(\cdot)$ implies the existence of $\delta_2 > 0$ and $L_2 > 0$ such that $Y^*(\cdot)$ is $L_2$-Lipschitz in $\mathcal{N}_{\delta_2}(x_1)$. Take $\delta = \min\{\delta_1, \delta_2, \delta_1/L_2\}$. The choice of $\delta$ ensures $(x_2, y_2) \in \mathcal{N}_{\delta_1}(x_1) \times \mathcal{N}_{\delta_1}(Y^*(x_1))$ for any $x_2 \in \mathcal{N}_\delta(x_1)$ and $y_2 \in Y^*(x_2)$. For any $x_2 \in \mathcal{N}_\delta(x_1)$, we pick

$$y_1 \in \underset{y \in Y^*(x_1)}{\arg\min} f(x_1, y), \quad y_2 \in \underset{y \in Y^*(x_2)}{\arg\min} f(x_2, y).$$

The Lipschitz continuity of $f(\cdot, \cdot)$ in $\mathcal{N}_{\delta_1}(x_1) \times \mathcal{N}_{\delta_1}(Y^*(x_1))$ and the Lipschitz continuity of $Y^*(\cdot)$ in $\mathcal{N}_{\delta_2}(x_1)$ implies there exist $y_1' \in Y^*(x_1)$ and $y_2' \in Y^*(x_2)$ such that

$$\varphi(x_1) - \varphi(x_2) \leq f(x_1, y_1') - f(x_2, y_2) \leq L_1 \left( \|x_1 - x_2\| + \|y_2 - y_1'\| \right) \leq (L_2 + 1)L_1 \|x_1 - x_2\|.$$
$$\varphi(x_2) - \varphi(x_1) \leq f(x_2, y_2') - f(x_1, y_1) \leq L_1 \left( \|x_1 - x_2\| + \|y_1 - y_2'\| \right) \leq (L_2 + 1)L_1 \|x_1 - x_2\|.$$

hold for any $x_2 \in \mathcal{N}_\delta(x_1)$, implying the locally Lipschitz property of $\varphi(\cdot)$.

**d.** We again use the function $f(x, y)$ in the proof of **b.** to bound (I) and (II) defined in Equation 8. Let $f(x, y) = -\min_{y_1 \in Y^*(x_1)} \|y - y_1\|$, then there exist $\delta_1 > 0$ and $L_1 > 0$ such that

$$(\text{I}) = \varphi(x_1) - \varphi(x_2) \leq L_1 \|x_1 - x_2\|, \quad \forall \|x_1 - x_2\| \leq \delta_1.$$

Let $f(x, y) = \max_{y_1 \in Y^*(x_1)} \|y - y_1\|$, then there exist $\delta_2 > 0$ and $L_2 > 0$ such that

$$(\text{II}) \leq \varphi(x_2) - \varphi(x_1) \leq L_2 \|x_1 - x_2\|, \quad \forall \|x_1 - x_2\| \leq \delta_2.$$

Together, taking $\delta = \min\{\delta_1, \delta_2\}$ and $L = \max\{L_1, L_2\}$ and recalling the definition of (I) and (II) in Equation 8, we can show that there exists some $\delta > 0$ such that it holds

$$\text{dist}(Y^*(x_1), Y^*(x_2)) \leq L \|x_1 - x_2\|, , \quad \forall \|x_1 - x_2\| \leq \delta,$$

which implies the local Lipschitz property of $Y^*(\cdot)$.

$\square$

## F  PROOFS IN SECTION 5

**Proposition 5.1.** *Under Assumption 5.1 or 5.2, $Y^*(x)$ is $(L/\alpha)$-Lipschitz. Furthermore, if $f(x, y)$ is $C_f$-Lipschitz, then $\varphi(x)$ is $(L/\alpha + 1)C_f$-Lipschtz.*

*Proof.* We show that $Y^*(x)$ is Lipschitz, and then $\varphi(x)$ is also Lipschitz by Proposition 4.3.

Under Assumption 5.1, for any $y_1 \in Y^*(x_1)$, there exists $y_2 \in Y^*(x_2)$ such that

$$\begin{aligned} &\alpha \|y_1 - y_2\| \\ &\leq \|\mathcal{G}_{1/L}(y_1; x_2) - \mathcal{G}_{1/L}(y_1; x_1)\| \\ &= L \left\| \mathcal{P}_{\mathcal{Y}} \left( y_1 - \frac{1}{L} \nabla_y g(x_2, y_1) \right) - \mathcal{P}_{\mathcal{Y}} \left( y_1 - \frac{1}{L} \nabla_y g(x_1, y_1) \right) \right\| \end{aligned}$$

$$\leq \|\nabla_y g(x_2, y_1) - \nabla_y g(x_1, y_1)\|$$
$$\leq L\|x_1 - x_2\|,$$

where we use $\mathcal{G}_{1/L}(y_1; x_1) = 0$ (Drusvyatskiy & Lewis, 2018) and Assumption 5.1 in the second line; the third line follows from the definition of the generalized gradient; the fourth line uses the non-expansiveness of projection operator by Corollary 2.2.3 in Nesterov (2018); and the last line uses the smoothness property of the lower-level function.

Under Assumption 5.2, for any $y_1 \in Y^*(x_1)$, there exists $y_2 \in Y^*(x_2)$ such that

$$2\alpha\|y_1 - y_2\|$$
$$\leq g(x_2, y_1) - g(x_2, y_2)$$
$$\leq g(x_1, y_1) - g(x_1, y_2) + 2L\|x_1 - x_2\|$$
$$\leq 2L\|x_1 - x_2\|,$$

where the last line uses $g(x_1, y_2) \leq g(x_1, y_2)$. $\qquad \square$

## G   Proofs in Section 6.1

First of all, we prove that the proposed Switching (sub)Gradient Method (SGM) in Algorithm 1 can find an LL optimal solution under the following Hölderian error bound condition.

**Assumption G.1.** *We suppose the lower-level function $g(x, \cdot)$ satisfies the $r$-th order Hölderian error bound condition on set $\mathcal{Y}$ with some coefficient $\nu > 0$, that is,*

$$\frac{\nu}{r}\text{dist}(y, Y^*(x))^r \leq g(x, y) - g^*(x), \quad \forall y \in \mathcal{Y}.$$

Note that this condition is also used by Jiang et al. (2023) and they show the following result.

**Lemma G.1** (Proposition 1 (i) in Jiang et al. (2023))**.** *Suppose that Assumption G.1 holds, $f(x, \cdot)$ is convex and $f(x, y)$ is $C_f$-Lipschitz. If a point $y$ satisfies*

$$f(x, y) - \varphi(x) \leq \zeta, \quad g(x, y) - g^*(x) \leq \frac{\nu}{r}\left(\frac{\zeta}{C_f}\right)^r, \tag{9}$$

*then we have $|f(x, y) - \varphi(x)| \leq \zeta$.*

Equation 9 can be achieved by the SGM, then we can show the following result for finding an LL optimal solution the Hölderian error bound condition.

**Theorem G.1.** *Under Assumption 6.1 and G.1, if we let*

$$\theta = \min\left\{\zeta, \frac{\nu}{4r}\left(\frac{\zeta}{C_f}\right)^r\right\}, \quad K_0 = K = \left\lceil \frac{4R^2\max\{C_f^2, C_g^2\}}{\theta^2} \right\rceil, \quad \tau = \frac{R}{\max\{C_f, C_g\}\sqrt{K}}. \tag{10}$$

*then Algorithm 1 can output a point $y_{\text{out}}$ satisfying $|f(x, y_{\text{out}}) - \varphi(x)| \leq \zeta$ within $\mathcal{O}\left(\frac{r^2\max\{C_f^2, C_g^2\}C_f^{2r}R^2}{\nu^2\zeta^{2r}}\right)$ first-order oracle complexity.*

*Proof.* Combine Lemma G.1 and Lemma G.3. $\qquad \square$

Next, we prove Lemma G.3, which relies on the following standard lemma for subgradient descent.

**Lemma G.2** (Subgradient Descent)**.** *Suppose $h$ is a $L$-Lipschitz convex function. For any $y, z \in \mathcal{Y}$, if we let $y^+ = \mathcal{P}(y - \tau\partial h(y))$, then it holds that*

$$h(y) - h(z) \leq \frac{1}{2\tau}(\|y - z\|^2 - \|y^+ - z\|^2) + \frac{\tau L^2}{2}.$$

*Proof.* See Theorem 3.2 in Bubeck et al. (2015). $\qquad \square$

Using this lemma, we then show the following result.

**Lemma G.3.** *Under the setting of Theorem G.1, the output of Algorithm 1 satisfies:*
$$f(x, y_{\text{out}}) - \varphi(x) \leq \theta, \quad g(x, y_{\text{out}}) - g^*(x) \leq 4\theta.$$

*Proof.* By Theorem 3.2 in Bubeck et al. (2015), the initialization step ensures $\hat{g}^*(x) - g^*(x) \leq 2\theta$.

Pick any $y^*(x) \in \arg\min_{y \in Y^*(x)} f(x, y)$ and denote $C = \max\{C_f, C_g\}$.

According to Lemma G.2 we obtain

$$f(x, y_k) - \varphi(x) \leq \frac{1}{2\tau}(\|y_k - y^*(x)\|^2 - \|y_{k+1} - y^*(x)\|^2) + \frac{\tau C^2}{2}, \quad k \in \mathcal{I};$$

$$g(x, y_k) - g^*(x) \leq \frac{1}{2\tau}(\|y_k - y^*(x)\|^2 - \|y_{k+1} - y^*(x)\|^2) + \frac{\tau C^2}{2}, \quad k \notin \mathcal{I};$$

Combing them together yields

$$\frac{1}{K}\sum_{k \in \mathcal{I}} f(x, y_k) - \varphi(x) + \frac{1}{K}\sum_{k \notin \mathcal{I}} g(x, y_k) - g^*(x) \leq \frac{R^2}{2\tau K} + \frac{\tau C^2}{2} = \frac{RC}{\sqrt{K}}. \tag{11}$$

With Equation 11 in hand, it suffices to show the result.

Firstly, we show that $\mathcal{I} \neq \emptyset$, and thus $y_{\text{out}}$ is well-defined. Otherwise, we would have the following contradiction:

$$2\theta \leq \frac{1}{K}\sum_{k=0}^{K-1} g(x, y_k) - \hat{g}^*(x) \leq \frac{1}{K}\sum_{k=0}^{K-1} g(x, y_k) - g^*(x) \leq \frac{RC}{\sqrt{K}} \leq \frac{\theta}{2}.$$

Secondly, we show that the output will not violate the constraint too much by:

$$g(x, y_{\text{out}}) - g^*(x) \leq \frac{1}{|\mathcal{I}|}\sum_{k \in \mathcal{I}}(g(x, y_k) - \hat{g}^*(x)) + (\hat{g}^*(x) - g^*(x)) \leq 4\theta.$$

Thirdly, we show that $f(x, y_{\text{out}}) - \varphi(x) \leq \theta$. It is trivial when $\sum_{k \in \mathcal{I}} f(x, y_k) - \varphi(x) \leq 0$ since it is an immediate result of Jensen's inequality. Therefore we can only focus on the case when $\sum_{k \in \mathcal{I}} f(x, y_k) - \varphi(x) > 0$. In this case, we can show that $|\mathcal{I}| \geq K/2$, otherwise we would have

$$\theta < \frac{1}{K}\sum_{k \notin \mathcal{I}} g(x, y_k) - \hat{g}^*(x) \leq \frac{1}{K}\sum_{k \notin \mathcal{I}} g(x, y_k) - g^*(x) \leq \frac{RC}{\sqrt{K}} \leq \frac{\theta}{2},$$

which also leads to a contradiction. Hence we must have $|\mathcal{I}| \geq K/2$, therefore, we obtain

$$f(x, y_{\text{out}}) - \varphi(x) \leq \frac{1}{|\mathcal{I}|}\sum_{k \in \mathcal{I}} f(x, y_k) - \varphi(x) \leq \frac{2}{K}\sum_{k \in \mathcal{I}} f(x, y_k) - \varphi(x) \leq \frac{2RC}{\sqrt{K}} \leq \theta.$$

This completes our proof. $\square$

We want to use Theorem G.1 to prove Theorem 6.1. Their only difference between them is the assumption. The following proposition shows that both Assumption 5.1 and 5.2 imply Assumption G.1 when $g(x, y)$ is convex in $y$. Therefore, the function class studied in Theorem 6.1 is contained in the function class studied in Theorem G.1.

**Proposition G.1.** *If $g(x, \cdot)$ is convex, then either Assumption 5.1 or 5.2 implies Assumption G.1.*

*Proof.* According to Corollary 3.6 in Drusvyatskiy & Lewis (2018), Assumption 5.1 implies Assumption G.1 with any $\nu < \alpha$ under the convexity of $g(x, \cdot)$ For Assumption 5.2, it is clear that it is equivalent to Assumption G.1 with $r = 1$. $\square$

Now we can easily prove Theorem 6.1.

**Theorem 6.1.** *Fix an $x$. Under Assumption 6.1, Algorithm 1 with appropriate parameters can ouput a point $y_{\text{out}}$ satisfying $|f(x, y_{\text{out}}) - \varphi(x)| \leq \zeta$ and $g(x, y_{\text{out}}) - g^*(x) \leq \zeta$ with $\mathcal{O}(\text{poly}(1/\zeta))$ first-order oracle calls from $g$.*

*Proof.* Combine Theorem G.1 and Proposition G.1.

$\square$

# H PROOFS IN SECTION 6.2

**Theorem 6.2.** *Given any $\varepsilon \lesssim C_f$. Let $\Delta = \varphi(x_0) - \varphi^*$. Under Assumption 6.2, set*

$$T = \mathcal{O}\left(d_x^{3/2}\left(\frac{C_\varphi^4}{\varepsilon^4} + \frac{\Delta C_\varphi^3}{\delta \varepsilon^4}\right)\right), \quad \eta = \Theta\left(\sqrt{\frac{\delta(\Delta + \delta C_\varphi)}{d_x^{3/2} C_\varphi^3 T}}\right). \tag{5}$$

*Then Algorithm 2 can output a point $x_{\text{out}}$ that satisfies $\mathbb{E}\min\left\{\|s\| : s \in \partial_\delta \varphi(x_{\text{out}})\right\} \leq \varepsilon$.*

*Proof.* First of all, we let

$$\varphi_\delta = \mathbb{E}_{v \sim \mathbb{P}_v}[\varphi(x + \delta v)],$$

where $\mathbb{P}_v$ is a uniform distribution on a unit ball in $\ell_2$-norm. Then, we define

$$\nabla_t = \frac{d_x}{2\delta}(\varphi(x_t + \delta u_t) - \varphi(x_t - \delta u_t))u_t. \tag{12}$$

According to Lemma D.1 in Lin et al. (2022), $\nabla_t$ satisfies the following properties:

$$\mathbb{E}_{u_t}[\nabla_t \mid x_t] = \nabla \varphi_\delta(x_t), \quad \mathbb{E}_{u_t}\left[\|\nabla_t\|^2 \mid x_t\right] \leq 16\sqrt{2\pi} d_x C_\varphi^2,$$

Then we know that

$$\mathbb{E}_{u_t}\left[\|\nabla_t - \hat{\nabla}_t\| \mid x_t\right] \leq \frac{d_x \zeta}{\delta}\mathbb{E}_{u_t}\|u_t\| = \frac{d_x \zeta}{\delta} \leq \frac{c_4 \varepsilon^2}{C_\varphi} \tag{13}$$

and

$$\begin{aligned}
&\mathbb{E}_{u_t}\left[\|\hat{\nabla}_t\|^2 \mid x_t\right] \\
&\leq 2\mathbb{E}_{u_t}\left[\|\nabla_t\|^2 \mid x_t\right] + 2\mathbb{E}_{u_t}\left[\|\nabla_t - \hat{\nabla}_t\|^2 \mid x_t\right] \\
&\leq 2\mathbb{E}_{u_t}\left[\|\nabla_t\|^2 \mid x_t\right] + \frac{2d_x^2 \zeta^2}{\delta^2}\mathbb{E}_{u_t}\|u_t\|^2 \\
&\leq 32\sqrt{2\pi} d_x C_\varphi^2 + \frac{2d_x^2 \zeta^2}{\delta^2} \\
&\leq c_1 d_x C_\varphi^2
\end{aligned} \tag{14}$$

for some positive constant $c_1, c_4 > 0$. Then we use the results of Equation 13, Equation 14 as well as the standard analysis of gradient descent to obtain

$$\begin{aligned}
\mathbb{E}\left[\varphi_\delta(x_{t+1}) \mid x_t\right] &\leq \varphi_\delta(x_t) - \eta\langle \nabla\varphi_\delta(x_t), \mathbb{E}[\hat{\nabla}_t \mid x_t]\rangle + \frac{c_2 \eta^2 C_\varphi \sqrt{d_x}}{2\delta}\mathbb{E}\left[\|\hat{\nabla}_t\|^2 \mid x_t\right] \\
&\leq \varphi_\delta(x_t) - \eta\|\nabla\varphi_\delta(x_t)\|^2 + \frac{\eta C_\varphi d_x \zeta}{\delta} + \frac{c_2 \eta^2 C_\varphi \sqrt{d_x}}{2\delta}\mathbb{E}\left[\|\hat{\nabla}_t\|^2 \mid x_t\right] \\
&\leq \varphi_\delta(x_t) - \eta\|\nabla\varphi_\delta(x_t)\|^2 + \frac{c_3 \eta^2 C_\varphi^3 d_x^{3/2}}{\delta} + \eta c_4 \varepsilon^2,
\end{aligned}$$

where we use Proposition 2.3 in Lin et al. (2022) that $\varphi_\delta$ is differentiable and $C_\varphi$-Lipschitz with the $(c_2 C_\varphi \sqrt{d_x}/\delta)$-Lipschitz gradient where $c_2 > 0$ is a positive constant and we define $c_3 = 2c_1 c_2$.

Telescoping for $t = 0, 1, \cdots, T$, we obtain

$$\mathbb{E}\|\nabla\varphi_\delta(x_{\text{out}})\|^2 \leq \frac{\Delta + \delta C_\varphi}{\eta T} + \frac{c_3 \eta C_\varphi^3 d^{3/2}}{\delta} + c_4 \varepsilon^2,$$

where we use $|\varphi_\delta(x) - \varphi(x)| \leq \delta C_\varphi$ for any $x \in \mathbb{R}^{d_x}$ by Proposition 2.3 in Lin et al. (2022).

Lastly, plugging the value of $\eta, T$ with a sufficiently small constant $c_4$ and noting that $\nabla\varphi(x_{\text{out}}) \in \partial_\delta \varphi(x_{\text{out}})$ by Theorem 3.1 in Lin et al. (2022), we arrive at the conclusion. $\square$