# OpenReview forum: "Bilevel Optimization without Lower-Level Strong Convexity from the Hyper-Objective Perspective"
_ICLR.cc/2024/Conference — Submitted to ICLR 2024_

### Official Review · Reviewer_V1pU · 2023-10-29

**Soundness:** 3 good
**Presentation:** 3 good
**Contribution:** 2 fair
**Rating:** 5
**Confidence:** 3

**Summary:**

In this paper, the authors focus on investigating bilevel problems that possess only Lower-Level (LL) convexity, without Lower-Level Strong Convexity (LLSC). This paper provides various hardness results to show the challenges of BLO without LLSC and further shows that the function satisfies either the gradient dominance condition or the weak sharp minimum condition, the hyper-objective is Lipschitz and thus is Clarke differentiable. Moreover, the authors find the $\zeta$-optimal solution to the LL problem via switching gradient methods within $\mathcal{O}(\text{poly}(1/\zeta))$ gradient oracle; and find the $(\delta, \epsilon)$-Goldstein stationary point of the hyper-objective via zeroth-order method within $\mathcal{O}(\text{poly}(1/\zeta, 1/\epsilon, 1/\delta))$ zeroth-order and first-order gradient oracle.

**Strengths:**

In general, the paper is well-written and clear. This paper explores several interesting topics related to BLO without LLSC:
1. Intractability of BLO without LLSC: The paper highlights that BLO without LLSC is generally considered intractable. This paper addresses this issue by presenting several interesting examples that highlight the difficulties associated with BLO without LLSC.
2. Impact of regularization: Regularization is indeed a common technique used to ensure the strong convexity of objective functions. However, the paper demonstrates through an example that regularization can have a significant impact on the hyper-objective, potentially leading to different objectives altogether. This emphasizes the importance of understanding how regularization affects the hyper-objective in the absence of strong convexity assumptions.
3. The paper introduces two sufficient conditions: the gradient dominance condition and the weak sharp minimum condition. These conditions serve as alternative criteria to ensure the continuity of the hyper-objective and the optimal solution of the lower-level function. By providing these conditions, the paper offers insights into overcoming the limitations of BLO without LLSC.

**Weaknesses:**

However, there are still important aspects that require attention or clarification.

- There is an important reference [1] that should pay more attention to:

  1. Reference [1] considers the bilevel problem with a general convex lower-level function, which is highly relevant to the current paper's investigation. However, this paper did not provide a detailed comparison with reference [1] in the related work section. Please clarify the main difference between your work and [1]. It would be beneficial to clarify the main differences between your work and [1] and highlight the contributions and unique aspects of your paper that distinguish it from the referenced work.

  2. According to your assessment, when the upper-level objective is convex, [1] requires  $\mathcal{O}(\max\left\lbrace1/\epsilon_f, 1/\epsilon_g \right\rbrace)$ iterations to find a solution that is $ \epsilon_f$ -optimal for the upper-level objective and $\epsilon_g$-optimal for the lower-level objective. However, based on my examination, the results presented in your paper are much worse compared to the results of [1]. It would be valuable to discuss and analyze the reasons for this discrepancy in the convergence performance between the two papers.
  3. Reference [1] also provides convergence complexity analysis under the Hölderian Error Bound, similar to the analysis presented in your paper's Appendix Section G. To provide a direct comparison between the two papers, it would be necessary to assess the specific convergence bounds, rates, or any additional insights derived in each paper.

  [1] Ruichen Jiang, Nazanin Abolfazli, Aryan Mokhtari, and Erfan Yazdandoost Hamedani. A conditional
          gradient-based method for simple bilevel optimization with convex lower-level problem. In AISTATS, 2023.


- This paper only claims that polynomial time algorithms for BLO with LL convexity. However, it does not provide the exact total gradient oracle complexity for Algorithms 1 and 2.
  1. What is the exact total gradient oracle complexity for the Algorithms 1 and 2?
  2. How is the complexity result compared to other related work?

**Questions:**

See the weaknesses above.

---

> ### Author Response · Authors · 2023-11-19
> **Responce to Reviewer V1pU**
>
> ### Weakness
>
> **W1: Comparison to CG-BiO**
>
> SGM can allow $f$ and $g$ to be nonsmooth, while CG-BiO requires them to be smooth. When both $f$ and $g$ are Lipschitz, SGM achives the rate of $\mathcal{O}(\max \{ 1/\epsilon_f^2,1/\epsilon_g^2 \})$. It is worth noticing that this rate of $\mathcal{O}(1/\epsilon^2)$ is optimal since it matches the lower bound for single-level nonsmooth optimization.
>
> CG-BiO has an improved complexity that breaks the nonsmooth lower bound by assuming smoothness of $f$ and $g$. Moreover, CG-BiO requires to solve a linear program at each iteration. Our algorithm only uses gradient descent steps, which is much simpler.
>
> **W2: Total complexity**
>
> The total complexity is
> $$
> \mathcal{O} \left( \left( \frac{d_x}{\delta \varepsilon^2} \right)^{2r} \frac{d_x^{3/2}}{\delta \varepsilon^4} \right),
> $$
> where $r=1$ for Assumption 5.2 and $r=2$ for Assumption 5.1.
>
> To the best of our knowledge, there are no other results showing non-asymptotic convergence to a UL stationary point without LLSC. Other works use other measures, and we think the complexity is not directly comparable.

---

### Official Review · Reviewer_Z4XB · 2023-10-31

**Soundness:** 2 fair
**Presentation:** 3 good
**Contribution:** 3 good
**Rating:** 6
**Confidence:** 3

**Summary:**

This paper presents a comprehensive analysis of the bilevel optimization without the typical  lower-level strong convexity (LLSC) assumption. Firstly, the hardness results showing the intractability of this kind of problem are given. Then, the paper introduces several key regularity conditions to make the problem tractable. Finally, the algorithm inexact Gradient-Free Method is proposed  and the corresponding non-asymptotic convergence is characterized.

**Strengths:**

(1) This paper gives non-asymptotic convergence to an upper-level stationary point for bilevel problem without LLSC, which is rarely studied.

(2) In addition to the theoretical analysis, the algorithm is also proposed to solve the problem.

(3) The theory is well-written. The examples are provided to help the readers have a better understanding of the theory.

**Weaknesses:**

The numerical results (only MSE and Test accuracy) are not convincing enough to support the convergence theory of the proposed algorithm.

(1) More details and analyses about the numerical results are recommended to present. For instance, it would be better to plot figures about the MSE/test accuracy with respect to the running time to have an intuitive understanding of the performance of the proposed algorithm.

(2) Furthermore, the authors are advised to test the toy example 5.2 in which the optimal solution is easy to calculate. Also, different numerical metrics ($|f-f^*|, |g-g^*|, \|x-x^*\|/\|x^*\|, \|y-y^*\|/\|y^*\|$) are recommended to plot to evaluate the convergence behaviors of the algorithms as in [1]. Besides, it is advised to compare with the recently proposed methods PBGD [2] and GALET [3] which are able to tackle the bilevel problem without LLSC.

[1] Risheng Liu, Pan Mu, Xiaoming Yuan, Shangzhi Zeng, and Jin Zhang. A generic first-order algorithmic framework for bi-level programming beyond lower-level singleton. In ICML, 2020.

[2] Han Shen, Quan Xiao, and Tianyi Chen. On penalty-based bilevel gradient descent method. In ICML, 2023.

[3] Quan Xiao, Songtao Lu, and Tianyi Chen. A Generalized Alternating Method for Bilevel Optimization under the Polyak-Lojasiewicz Condition[J]. arXiv preprint arXiv:2306.02422, 2023.

**Questions:**

Since it is possible to encounter the bad $x_{out}$ that $\{\|s\|: s \in \partial_{\delta} \varphi(x_{out})\}$ is large, can you explain how to choose the output  in Algorithm 2 in detail in the experiment?

---

> ### Author Response · Authors · 2023-11-19
> **Responce to Reviewer Z4XB**
>
> ### Weakness:
>
> Thank you for your recognition of our work. We answer your questions below.
>
> > W1 and W2: More numerical results
>
> We are happy to add more experiment details.
> Due to the time limit in the rebuttal period, we make sure to add the experiments the reviewer suggests (including Toy example 5.2, comparison with PBGD and GALET, running time, etc. ) in the next revision.
>
> ### Questions
>
> > Q1: How to choose the output?
>
>    In practice, simply choosing the point in the last iteration works well.

---

### Official Review · Reviewer_f4Tk · 2023-10-31

**Soundness:** 2 fair
**Presentation:** 2 fair
**Contribution:** 2 fair
**Rating:** 3
**Confidence:** 2

**Summary:**

The authors propose an algorithm to solve bilevel optimization problems with non-strongly convex lower optimization problems.

**Strengths:**

Authors tackle a hard problem, without strong convexity, the lower optimization problem can be multi-valued, which makes the bilevel optimization problem much harder.

**Weaknesses:**

Clarity: what's the main result of the paper? I was not able to properly parse it. How realistic is assumption 2? Why does Theorem 6.2 require assumption 6.2 and Corollary 6.1.

The paper is an arid succession of definitions and propositions: I was not able to understand the proposed algorithm and results.
Since I did not understand the paper, I was not able to judge its quality. I think the paper requires a significative writing revision to be accessible for an ML audience.

**Questions:**

How do you choose $\tau$ in practice?

---

> ### Author Response · Authors · 2023-11-19
> **Responce to Reviewer  f4Tk**
>
> ### Weakness
>
> > What is the main result of this paper?
>
> Our hardness results show that bilevel optimization for general convex lower-level functions is intractable to solve. We then identify several regularity conditions of the lower-level problems that can provably confer tractability. Under these conditions, we propose the Inexact Gradient-Free Method (IGFM), which uses the Switching Gradient Method (SGM) as an efficient sub-routine, to find an approximate stationary point of the hyper-objective in polynomial time.
>
> > How realistic are the assumptions? And why they are needed.
>
> Assumption 6.1 has (a)-(e) assumptions. We explain more below.
>
> Assumption (a) is the convexity assumption, since we are required to solve the lower-level sub-problem, while finding a global solution of nonconvex problems is NP-hard.
>
> Assumptions (b)- (d) are regular conditions that can usually be satisfied.
>
> Assumption (e) is exactly the growth conditions we introduce. It ensures that the hyper-objective is continuous, Clarke differentiable and the lower-level problem can be solved. Therefore, we think this condition is very important.
>
> We also provide examples that satisfy our assumptions in Section 5.2.
>
> ### Questions
>
> > Q1 How to choose $\tau$ in practice?
>
> Tune this parameter in a grid or merely set $\tau$ to be some small number such that $0.001$.

---

### Official Review · Reviewer_y92G · 2023-11-01

**Soundness:** 2 fair
**Presentation:** 3 good
**Contribution:** 2 fair
**Rating:** 3
**Confidence:** 3

**Summary:**

The authors explore bilevel optimization (BLO) without relying on the typical lower-level strong convexity (LLSC) assumption, approaching it from the optimistic hyper-objective perspective. They demonstrate that solving BLO with general convex lower-level (LL) functions is computationally challenging. To address this, they introduce two regularity conditions for the LL problems, which can make BLO tractable even with only lower-level convexity. Additionally, they propose novel algorithms designed to find a LL optimal solution and an upper-level stationary point, with non-asymptotic convergence guarantees.

**Strengths:**

S1: Bilevel optimization is a timely topic. Investigating BLO without relying on the conventional lower-level strong convexity is a relatively unexplored area, and the topic addressed in this work is interesting and important.

S2: The paper is well-written and easy to follow. Examples 4.1 is helpful to understand the main result.

S3: A novel zeroth-order method for studying BLO without relying on the typical lower-level strong convexity is introduced.

**Weaknesses:**

W1: The paper lacks coverage of closely related papers, such as:

[1] Risheng Liu, Yaohua Liu, Wei Yao, Shangzhi Zeng, and Jin Zhang. Averaged method of multipliers for bi-level optimization without lower-level strong convexity. In ICML, 2023.

[2] Han Shen, Quan Xiao, and Tianyi Chen. On penalty-based bilevel gradient descent method. In ICML, 2023.

W2: Some of the results in this paper may not be competitive. For example,

(1) Compared to Prop. 4.1, the example in Remark 4 of [3] has shown that adding a regularization term to the LL function does not work well without LLSC. This example also support Prop. 4.1 and $\inf \varphi(x)$ is finite.

[3] R Liu, P Mu, X Yuan, S Zeng, J Zhang. A General Descent Aggregation Framework for Gradient-based Bi-level Optimization. In IEEE Transactions on Pattern Analysis and Machine Intelligence, 2022.

(2) Compared to SGM (i.e., Algorithm 1), the algorithm CG-BiO in Jiang et al. (2023) can find an LL optimal solution in polynomial time. And it seems that SGM does not have a better iteration complexity.

W3: I did not thoroughly review all the proofs of the main results, but I noticed some errors in certain proofs, including the proof of Proposition 4.2 (e.g., details in Q2). The authors should consider reviewing and verifying all their results for accuracy.

**Questions:**

Q1.Proposition 4.1: Prop 4.1 is used to illustrate that any small regularization to the LL function without LLSC may lead to a large deviation on the hyper-objective. But,

(1) The instance in the proof of Prop. 4.1 is not good enough since $\varphi(x)=-x^2$ and then $\inf \varphi(x)=-\infty$. Similarly, $\inf \varphi_{\lambda}(x)=-\infty$. Hence the conclusion of Prop. 4.1 is about $(-\infty)-(-\infty)$.

(2) The example in Remark 4 of [1] has shown that adding a regularization term to the LL function does not work well without LLSC. This example also support Prop. 4.1 and $\inf \varphi(x)$ is finite.

[1] R Liu, P Mu, X Yuan, S Zeng, J Zhang. A General Descent Aggregation Framework for Gradient-based Bi-level Optimization. In IEEE Transactions on Pattern Analysis and Machine Intelligence, 2022.

Q2: Proposition 4.2: In the proof of Prop. 4.2, the authors said that $\nabla f(x, y_k)=0$, which contradicts to the defintion of $f(x,y)=\frac{1}{\sqrt{K}}\sum_{j=K+1}^{q} y_{[j]}$. Please check the proof of Prop. 4.2. Another question is: does the function $g(x, \cdot)$ in the proof of Prop. 4.2 is strongly convex since $d_y=q=2K$?

Q3: Example 4.1: What is the defition of $sign(0)$? If sign(0)=0 in the classical sense, then $\varphi(0)\neq 0^2+sign(0)$ since $\varphi(0)=-1$.

Q4: SGM, Algorithm 1: Compared with CG-BiO in Jiang et al. (2023), what is the advantage of using SGM to find an LL optimal solution? Please compare SGM with CG-BiO in more detail.

Q5: The sentence following Corollary 6.1: Why do you say “it is the first theoretical analysis that shows the non-asymptotic convergence to a UL stationary point for BLO without LLSC” ? As I know, there are at least two related papers as follows:

[1] Risheng Liu, Yaohua Liu, Wei Yao, Shangzhi Zeng, and Jin Zhang. Averaged method of multipliers for bi-level optimization without lower-level strong convexity. In ICML, 2023.

[2] Han Shen, Quan Xiao, and Tianyi Chen. On penalty-based bilevel gradient descent method. In ICML, 2023.

Q6: It is better to compare IGFM with the algorithms in the above two closely related papers.

Minor comments:

(1) Proposition 4.1: $g_{\lambda}(x,y)$ should be $g(x,y)$.

(2) Assumption 5.1: $L_g$ should be $L$ and some norm of $\mathcal{G}_{1/L}(y;x)$ is missed.

(3) Example 5.3: check the Lipschitz constant $L$ and the constant $\alpha$.

(4) Example 5.4: $d_y=n$.

---

> ### Author Response · Authors · 2023-11-19
> **Responce to Reviewer y92G**
>
> ### Weakness
>
> > W1: The paper lacks coverage of closely related papers
>
> We are happy to cite these additional related works. However, Liu et al. and Shen et al. show no convergence to UL stationary points (Definition 3.8). They use different stationary measures.
>
>
> > W2: Some of the results in this paper may not be competitive.
>
> (1) Thanks for pointing out Remark 4 from Liu et al. We did not realize this when writing our paper.
> We are happy to replace  Proposition 4.1 with the result of Liu et al. in the main text.
>
> (2) Comparison with CG-BiO.
>
> SGM can allow $f$ and $g$ to be nonsmooth, while CG-BiO requires them to be smooth. When both $f$ and $g$ are Lipschitz, SGM achives the rate of $\mathcal{O}(\max\{1/\epsilon_f^2,1/\epsilon_g^2 \})$. It is worth noticing that this rate of $\mathcal{O}(1/\epsilon^2)$ is optimal since it matches the lower bound for single-level nonsmooth optimization.
>
> CG-BiO has an improved complexity that breaks the nonsmooth lower bound by assuming smoothness of $f$ and $g$. Moreover, CG-BiO requires to solve a linear program at each iteration. Our algorithm only uses gradient descent steps, which is much simpler.
>
> > W3：Proof of Proposition 4.2
>
>  We apologize for the incorrectness in the initial version. We have fixed it with a slight modification of proof (marked by red color) in the new PDF.
>
>
> ### Questions
>
> > Q1: Proposition 4.1
>
> Thank you again. We answer this question in W2(1).
>
> > Q2: Proof of Proposition 4.2.
>
> Thank you again for pointing out the mistake. We answer this question in W3.
>
> Nesterov's worst-case function for convex optimization is strictly convex, and we use the same function.
>
> > Q3: Example 4.1
>
> Thanks a lot. We will replace ${\rm sign}(x)$ with $I[x \le 0]$ to make it more clear.
>
> > Q4: Comparison to CG-Bio
>
> We answer this question in W2(2).
>
> > Q5 and Q6. It is the first theoretical analysis that shows the non-asymptotic convergence to a UL stationary point for BLO without LLSC.
>
>
> We answer this question in W1.
>
> ### Minor Commnents
>
> Thank you so much for pointing out the typos. We will fix them.

---

### Official Review · Reviewer_9BLf · 2023-11-01

**Soundness:** 3 good
**Presentation:** 3 good
**Contribution:** 3 good
**Rating:** 5
**Confidence:** 3

**Summary:**

This paper considers the bilevel problem without strongly convex assumption. It provides the reason why the BLO without LLSC is generally intractable. Based on the analysis, this paper provides novel polynomial time algorithms for BLO with LL convexity and experimental results show the superiority of the proposed method.

**Strengths:**

1.	This paper is well-written and theoretical solid.
2.	This paper explains the difficulty of the BLO without LLSC is important.
3.	This paper evaluates the proposed method on two different applications.

**Weaknesses:**

1.	I think the proposed algorithm. 2 is similar to that proposed in [1].
2.	Algorithm 2 needs to solve the lower-level problem. I think this will make the proposed method much slower than the result shown in Theorem 6.2.
3.	This paper claims that it is the first paper show the non-asymptotic convergence to a UL stationary point for BLO without LLSC. However, [2] and [3] provide better results in convergence rate.

[1] Gu B, Liu G, Zhang Y, et al. Optimizing large-scale hyperparameters via automated learning algorithm[J]. arXiv preprint arXiv:2102.09026, 2021.
[2] Huang F. On momentum-based gradient methods for bilevel optimization with nonconvex lower-level[J]. arXiv preprint arXiv:2303.03944, 2023.
[3] Daouda Sow, Kaiyi Ji, Ziwei Guan, and Yingbin Liang. A constrained optimization approach to bilevel optimization with multiple inner minima. arXiv preprint arXiv:2203.01123, 2022.

**Questions:**

1.	Can authors provide the results on some lower-level constrained BLO?
2.	Why Algorithms contains $K_0$, $K$, $\theta$?
3.	Does Theorem 6.2 take the time of solving the inner problem into account?
Other questions shown in weakness.

---

> ### Author Response · Authors · 2023-11-19
> **Responce to Reviewer 9BLf**
>
> ### Weakness
>
> > W1: Comparison to Gu et al.
>
>
> We thank the reviewer for bringing this paper to our attention.
>
> Our work differs from Gu et al. in the following aspects.
>
> 1. **Problem Setup**. We focus on bilevel optimization, while the problem in Gu et al. is a composition optimization. Our problem is more challenging.
>
> 2. **Motivation of Introducing Zeroth-Order Methods**. We introduce zeroth-order methods for theoretical purpose: it can smooth the hyper-objective which we found can be nonsmooth for BLO without LLSC, and show non-asymptotic convergence to a Goldstein stationary point.
> Gu et al. propose zeroth-order for practical consideration. This work is complementary to ours. And we are happy to cite the reference.
>
> > W2: Needs to solve the lower-level problem
>
> Algorithms under LLSC also require solving lower-level problems.
>
> > W3: Comparison to Sow et al. and Huang et al.
>
> Sow et al. did not show convergence to UL stationary points. Instead, they show convergence to KKT points, which may not be a necessary condition. See Appendix A for detailed discussions.
>
> Huang et al. suppose that the lower-level solution set is a singleton. Our problem is much more challenging since we allow the lower-level solution set not to be singleton.
>
> ### Questions
>
> > Q1: Can authors provide the results on some lower-level constrained BLO?**
>
> We have provided results on lower-level constrained BLO. Theorem 6.1 and 6.2 work for both unconstrained and constrained cases.
>
> > Q2:Why Algorithms contains $K_0,\tau,\theta$?
>
> They are hyper-parameters for the algorithm. Their meanings are clear from the procedure of the algorithm.
>
> > Q3: Does Theorem 6.2 take the time of solving the inner problem into account?**
>
> Corollary 6.1 takes this into account.

---

### Official Review · Reviewer_ShQk · 2023-11-06

**Soundness:** 2 fair
**Presentation:** 2 fair
**Contribution:** 2 fair
**Rating:** 3
**Confidence:** 3

**Summary:**

The authors consider bilevel optimization problems and propose to relax strong convexity conditions for the lower level problem. The authors propose a uniform version of the gradient growth condition (or error bound condition, or PL condition, those are essentially equivalent in the convex setting).

**Strengths:**

Pushing implicit differentiation beyond strongly convex lower level problems is of great relevance and uniform growth condition is indeed a natural candidate.

**Weaknesses:**

The proposed discussion is too weak to constitute a relevant contribution for ICLR:
- No non-trivial example: the only explicit bi-level problems which relax lower level strong convexity are quadratic programs on linear equality constraints which can be solved by linear algebra.
- Section 4 is of little use, it describes known issues related to discontinuity of solution sets and contains incorrect statements.

**Questions:**

## Absence of discussion and examples on the proposed growth condition

Section 5 is nice, I think it is really the "heart" of the idea of relaxing strong convexity. However I think that it is too abstract as the authors do not provide concrete examples of bilevel problems where this condition is satisfied. The PL condition is known to be a good relaxation of strong convexity, but the authors ask that it holds uniformly (with a fixed alpha) for all x. This is a strong requirement and I wonder if there are many problems which satisfy this. Examples 5.3 and 5.4 are nice but the lower problem is quadratic which means that the optimality condition is equivalently expressed with an affine constraint, since in both cases, the set Y^*(x) can be described using a linear constraint on (x,y). In 5.4, the lower level problem is reduced to lnear equality constraints $M y = \frac{y_0 - Ax}{\lambda - 1}$ (if I did not make any mistake). As a result, both bilevel examples reduce to quadratic objective under linear equality constraint which reduces to a linear system (first proposed "Solution method" on wikipedia's quadratic programming page).

Since this is the heart of the paper, I think that it deserves more example and a more detailled discussion about how restrictive is Assumption 5.1.

This is important for further theoretical results as Theorem 6.2 requires to know alpha (or the Lipschitz constant of $\phi$ which is similar), and therefore the only examples provided in the paper for which Theorem 6.2 applies either have strongly convex lower level, or the lower level solution reduces to linear constraints and constitute easy instance of bilievel programming.

Besides this absence of discussion is problematic since the numerical section 7.1 is only based on 8 dimensional occurences of such examples which can be globally solved very efficiently using numerical linear algebra. Section 7.2 is a logistic regression problem for which the lower level is locally strongly convex.

Finally, it would be nice to have a comparison with the parametric Morse-Bot condition of Arbeel and Mairal.

A similar comment holds for assumption 7.2, is there a general class of problems for which it is satisfied?

## Sections 4

I do not think that Proposition 4.1 is usefull, infinity means that one of the problems has no solution, which is somewhat degenerate. What is the purpose of this proposition regarding the message of the paper? The proof is not understandable at all: what is f what is g? This essentially illustrates discontinuity of solution sets which induce discontinuous jumps in a bilevel context. Here this is obvious due to the fact that the lower level does not depend on y1, so adding regularization creates a discontinuity for $\lambda = 0$. This is a known phenomenon (see comment on Section 4.3) below.

Proposition 4.2 is litterally false, the variable k should be quantified (for all k) and I guess it should be related to K (for all k leq K). Furthermore it is essentially a static result as x is fixed, which illustrates the fact that first order methods may produce iterates far from the solution for a large number of steps (this is known and as the authors acknowledge, this is presented for example in Nesterov's textbook). It is seen in the proof: x plays no role here.

The proof of Proposition 4.2 is false and not understandable. The curcial step is the equation before "therefore f(x,y_k) remains zero". Since f is linear, \nabla f is a constant (and nonzero due to the definition of f) so \nabla f = 0 is false. Furthermore y_{k,[j]}=0 is also false and the conclusion "therefore f(x,y_k) remains zero", is false. Indeed, in the authors example, y_0 + span( \nabla_y f(x,y_0)) contains elements y such that f(x,y) = 1.

Section 4.3 is known: for example Dontchev and Rockafellar have a whole book section dedicated to regularity properties of solution mappings (Implicit Functions and Solution Mappings A View from Variational Analysis, chapter 3), precisely because solution mappings are discontinuous. Proposition 5.3 is nice but it is also well known that continuity properties of the solution mapping translate to continuity of the bilevel value function: see e.g. theorem 3B.5 in Dontchev Rockafellar which contains a great deal of the ideas exposed in appendix D.

## Minor comments
- Remark 3.1: "In Definition 3.8, we assume that φ(x) is locally Lipschitz, which is one of the mildest conditions to ensure Clarke differentiability". This is indeed the case for the proposed definition of Clarke subdifferential, but this is a special case of a more general definition given by Clarke which allows to model discontinuous functions as well. This remark is essentially false

- Definition 3.7, technically, g(x,y) is not a function.

- In the numerical section: what is an epoch? Is the computation time comparable to number of epochs for each algorithm?

---

> ### Author Response · Authors · 2023-11-19
> **Responce to Reviewer ShQk**
>
> We greatly appreciate the feedback from the reviewer, and we find the suggestions very helpful in improving the quality of our articles.
>
> > Q1: Absence of discussion and examples on the proposed growth condition
>
> **More examples**
>
> One good example of Assumption 5.1 is neural network training in the NTK (neural tangent kernel) region. It can be proved that it satisfies the PL condition.
>
> **Comparison with the Morse-Bot condition**
>
>    The Morse-Bott condition implies the PL condition locally by Arbeel and Mairal. Therefore, our Assumption 5.1 recovers the PL condition in the unconstrained case. Therefore, we think our assumption is more general.
>
> > Q2: Section 4
>
> **Purpose of Proposition 4.1**
>
> This proposition shows that adding an arbitrarily small regularization on the LL function may change the hyper-objective $\varphi(x)$ from a quadratic to a linear function.
>
>
> **Proof of Proposition 4.2.**
>
>    We apologize for the incorrectness in the initial version. We have fixed it with a slight modification of proof (marked by red color) in the new PDF.
>
> **Section 4.3 is known.**
>
>    We thank the reviewer gratefully for pointing out  Theorem 3B.5 in Dontchev Rockafellar. We will cite this in the future version.
>
>    It is worth noticing that our Proposition goes further. Besides the **"if"** direction by Dontchev Rockafellar,  we also show the **"if and only if"** direction. To prove this "conversely part" (Part b and d in Proposition 5.3), we construct some functions to achieve this, which is of independent technical interest. We find no reference showing this prior to us. And we think our proposition is still valuable.
>
>
> > Minor comments
>
> **Remark 3.1** Thanks a lot. We changed "one of the mildest conditions" to "a regular condition" in the new PDF (marked in red color in supplementary).
>
> **Meaning of "epoch" in the experiment.**
>     "epoch" means the number of iterations in the upper-level variable $x$. The number of epochs is proportional to the running time since we fix the same number of lower-level iterations in $y$.

---

> > ### Comment · Reviewer_ShQk · 2023-11-22
> >
> > I have read the rebutal of the authors and I keep my score unchainged.
> >
> > To elaborate on the assumptions and the proposed growth condition: it is not the PL condition itself that should be discussed, but the fact that it holds uniformly (with a uniform constant) with respect to the parameters. This is a very strong requirement.

---

### Meta-Review · Area_Chair_PA6V · 2023-12-05

**Metareview:**

This paper tackles the issue of developing algorithms to solve bilevel optimization without strong convexity assumption on the inner level. Some of the current litterature on BO is missing (highlighted by several reviewers) The rebuttal lacks discussion on some important points raised by reviewers (especially ShQk). Asking a function to be uniformly PL is incredibly demanding (and is at the core of the paper)!

**Justification For Why Not Higher Score:**

Lack of literature review / very strong assumption

**Justification For Why Not Lower Score:**

N/A

---

### Decision · Program_Chairs · 2024-01-16

Reject